# Soft magnetic microrobots with remote sensing and communication capabilities

Quan Gao [1,7], Minsoo Kim [1,7] ✉, Denis von Arx[1], Elric Zhang[1], Xinzhi Zhang[2], Hao Ye [1], Christian Vogt [3], Claas Ehmke [1], Dianne Corsino[4], Federica Catania[4], Niko Münzenrieder [4], Michele Magno[3], Giuseppe Cantarella [5,6], Bradley J. Nelson [1] & Salvador Pané [1]

Remote communication in small-scale robotics offers a powerful way to enhance their capabilities, introducing options for state monitoring, multi-agent collaboration, and autonomous operation. Integrating common remote communication tools, such as antennas, into microrobots is challenging with conventional design and manufacturing techniques. We propose a concept that integrates shape-reconfigurable soft microrobots with flexible electronics, leveraging their elastic mechanical properties to enable remote communication. This approach, based on photolithography processes, is scalable and adaptable to various sensing applications. As a proof of concept, we present a microrobot, which integrates a thermoresponsive magnetic hydrogel, an anisotropic support structure, and a flexible dipole antenna into a cohesive three-layered design. The microrobot can morph from a helical shape at low-temperatures to a planar shape at high-temperatures. This shape transformation can be remotely detected by external radio communication receivers, enabling shape-state recognition and environmental temperature sensing. Furthermore, we show that the collective behavior of multiple microrobots enhances the recognition performance by amplifying the signal. The concept represents a significant advancement in co-engineering smart materials and flexible electronics, illustrating an approach of microrobotic embodied intelligence by integrating environmental monitoring, magnetic navigation, and remote signaling.

Microorganisms, despite their microscopic size, exhibit robust intelligence that allows them to thrive in complex environments through adaptation and communication. For example, *Escherichia coli* uses membrane chemoreceptors to detect amino acids, adjusting flagellar rotation to navigate toward nutrient sources[1]. Additionally, through a communication process called quorum sensing, these bacteria monitor population density and exchange chemical signals to coordinate collective behaviors and regulate gene expression dynamically[2]. Certain bacteria can also respond to physical cues such as temperature and magnetic fields through thermotaxis or magnetotaxis[3].

Inspired by these natural strategies, roboticists have incorporated principles of microscale intelligence into microrobotic designs[4]. A central approach involves using functional materials that respond to varied stimuli, allowing microrobots to sense, process information,

[1]Multi-Scale Robotics Lab, Institute of Robotics and Intelligent Systems, ETH Zurich, Zurich, Switzerland. [2]Institute of Electromagnetic Fields, ETH Zurich, Zurich, Switzerland. [3]Center for Project-Based Learning, ETH Zurich, Zurich, Switzerland. [4]Faculty of Engineering, Free University of Bozen-Bolzano, Bozen-Bolzano, Italy. [5]Department of Physics, Informatics and Mathematics, University of Modena and Reggio Emilia, Modena, Italy. [6]Istituto Nanoscienze CNR, Centro S3, Modena, Italy. [7]These authors contributed equally: Quan Gao, Minsoo Kim. ✉e-mail: minkim@ethz.ch

and navigate autonomously[5–7]. Despite the challenges associated with micro- and nanoscale fabrication, significant progress has already been made in developing intelligent microrobotic systems, ranging from soft[7–11], to compound[12], and reconfigurable designs[13–20], as well as encodable[17,19,21–25], multifunctional[17,26,27], and integrated systems[28,29]. By incorporating anisotropic magnetic designs, researchers have introduced improved navigation and previously unobtainable functionalities into versatile microrobotic platforms[8,17,19,30]. These individual microrobots have further advanced to multiscale[12], multimodal[14,15], multi-agent[31], hierarchical[32,33], self-organizing[22,34], and swarm behaviors[13,14,35,36]. And in doing so, they have expanded their applications in environmental remediation[37], micromanipulation[38], medicine[39,40], and sensing[41] contexts.

An important aspect of microscale intelligence focuses on communication and collective behavior. Through communication, simple local interactions can develop into elaborate complex group interactions, enabling sophisticated patterns and actions. This coordinated cooperation allows for the design of microrobots that are less complex individually but more skilled as a collective. Additionally, microrobots can further benefit from additional advantages over bacteria external agents, as their communication with external agents can enhance localization, environmental monitoring, and autonomy.

The integration of these technologies into microrobots became possible in the last few decades due to advances in microfabrication techniques, including those commonly used in microelectromechanical systems (MEMS)[42–44] and complementary metal-oxide-semiconductor (CMOS)[28,29,45], alongside emerging methods such as 3D direct laser writing[7,24,27,46]. For example, Bandari et al. used microfabrication to create inductively powered microrobots with a strain engineering-based form of chemical propulsion[29], and Miskin et al. integrated photovoltaics and surface electrochemical actuators for a light-powered microrobot[28]. While advanced fabrication techniques have shown up possibilities for actuation methods, these implementations still struggle with a critical capability for real, practical utility. Namely, these microrobots cannot communicate real-time information about their environment with external systems.

Roboticists have attempted to solve this problem in numerous fashions, whether by incorporating commercial antenna chips (including RFID) or through tethered approaches. These methods, based on readily available, low-cost, and robust transmission technologies, can relay important information about their environment to external agents. For example, Li et al. demonstrated radiofrequency (RF)-capable, magnetically navigated microrobots that wirelessly transmitted temperature and pH data[47]. However, both the chips and required auxiliary circuits force fundamental tradeoffs between signal quality and microrobot size while also requiring a rigid structure. Han and colleagues addressed some of these issues by using microfabrication and compressive buckling techniques to incorporate a flexible electronic sensor array within a balloon catheter that measured pressure and temperature[48]. These sensing functionalities can be enhanced using various flexible and stretchable electronic systems[49–53], as these devices offer high mechanical compliance and multi-functionality through the integration of thin, cellular-scale semiconductor components. The soft nature of the flexible system better matches the mechanical properties seen in biology, but the setup still restricts the working range by tethered design, and the wireless alternatives require substantial external RF power transmitters. As such, the question remains whether an untethered can successfully integrate the key aspects of remote navigation, collective behavior, and wireless sensing into a single system while maintaining the key advantage of flexible soft microrobots.

This research seeks to answer that question with a microrobotic approach that integrates flexible electronics and a shape-reconfigurable soft microrobot into a single device. This leverages smart materials to create large, physical changes in the microrobot in response to local stimuli and a flexible electronic antenna design that can take advantage of the entire microrobotic surface to propagate its signal. By coupling these two together, the physical changes to the microrobotic structure produce equally dramatic changes in the antenna signals, enabling instant remote communication within a fully flexible microrobotic system. To achieve this, we developed an integration protocol to combine an anisotropic SU-8 passive layer with an iron oxide nanoparticle (IONP) embedded thermally-responsive hydrogel active layer, leading to a temperature-dependent helical-to-planar transformation function. Subsequently, we laminated a flexible dipole antenna for radio communication, completing the microrobot. This microrobot demonstrated magnetic navigation, RF communication-based shape detection, localization, and remote temperature sensing. The collective behavior of the multiple microrobots showed both enhanced RF signal sensitivity and improved reliability.

## Results

### Microrobots capable of shape perception

The microrobot consists of three main layers, as shown in Fig. 1a. The first corresponds to the flexible dipole antenna layer. The antenna is realized on an anisotropic support layer, which is attached to a thermoresponsive hydrogel composite layer comprising a poly-N-isopropylacrylamide (pNIPAM) with embedded iron oxide nanoparticles (detailed information on the layer design and parameters provided in the subsequent section). At low temperatures, the thermoresponsive hydrogel forms a porous mesh, allowing water molecules to penetrate the network, causing it to swell. This volumetric change in the hydrogel layer induces bending moments in layered structures (Supplementary Fig. 1a and Movie S1) and interacts with the anisotropic supporting layer to generate torque for twisting (Supplementary Fig. 1b). This results in a three-dimensional helical structure (Fig. 1b), optimized for magnetically guided corkscrew motion. The three-layer helical structure is shown in Fig. 1c.

Figure 1d provides an overview of the microrobotic concept and its intended use case. In its hydrated state, within a standard temperature range state, the microrobot's helical shape enables propulsion and guidance through fluidic environments via rotating magnetic fields. When the environmental temperature exceeds 40 °C, surpassing the phase change threshold, the thermoresponsive hydrogel dehydrates and shrinks volumetrically, causing the microrobot to morph from a helical to a planar shape. As the microrobot changes shape, the laminated dipole antenna deforms accordingly (Supplementary Fig. 1c, d), altering its RF signal response in a way that can be detected by an external agent.

### Flexible electronics and soft microrobot integration processes and layered structure design

Figure 2a outlines the process used to integrate flexible electronics into soft microrobots, with three key fabrication steps for the electronics layer, the passive layer, and the active layer (details provided in the "Methods" section). Initially, the electronics (i.e., the dipole antenna) are patterned onto the sacrificial layer (polymethyl methacrylate, PMMA)-coated silicon wafer (Fig. 2b). Next, the passive layer is formed by spin-coating SU-8 onto the electronics layer and structuring the rectangular and scaffold patterns via photolithography (Fig. 2c). Spacers are then added around the structures to provide room for the hydrogel composites to fill the space between the upper cover glass (i.e., photomask) and the lower silicon wafer. A photomask coated with a sacrificial layer (polyvinyl alcohol, PVA) is aligned with the spacer, forming a sandwich structure. Note that alignment markers are designed to precisely position the layered designs, including the electronics, scaffold structures, spacers, and photomask patterns (Supplementary Fig. 2). To ensure a strong bonding and prevent delamination during hydrogel swelling (Supplementary Movie S2), the SU-8 surface requires treatment after being patterned. Finally, the

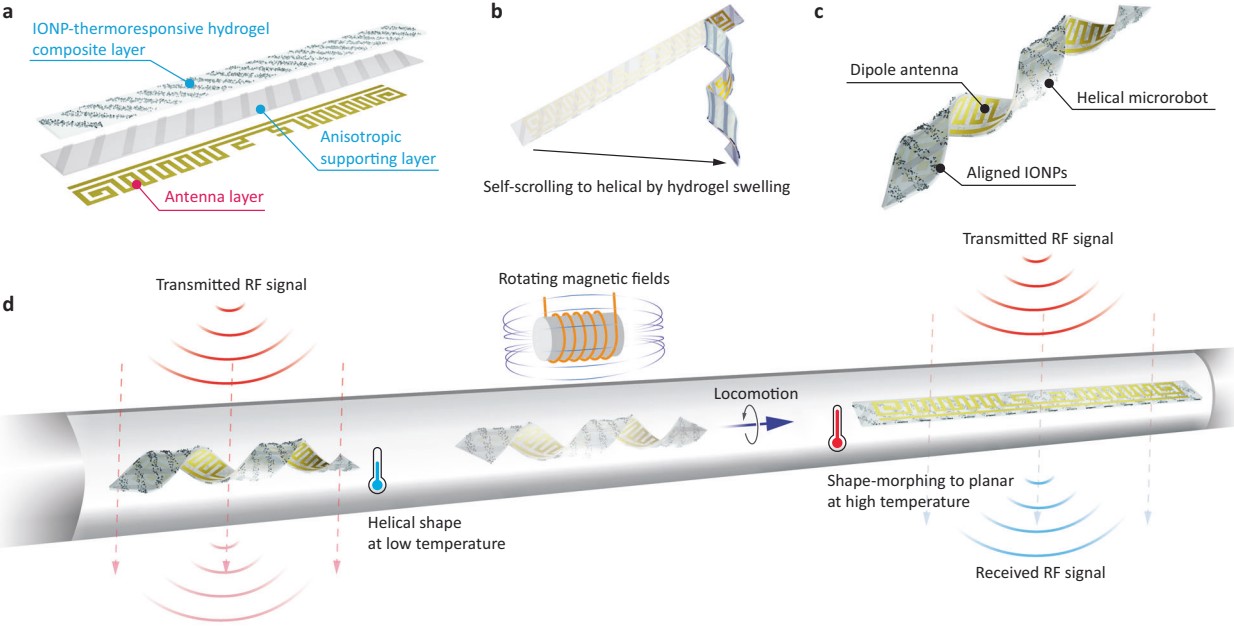

**Fig. 1 | Integration of flexible electronics and soft microrobots for remote shape-sensing microrobots. a** Three-layer configuration of the integrated soft microrobot and flexible electronics. The active layer employs iron oxide nanoparticle (IONP)-thermoresponsive hydrogel composites. The second passive layer consists of SU-8 supports and scaffolding, while the final electronic layer forms the antenna. **b** Helical shape formation by self-scrolling. Swelling of a hydrogel layer induces bending moments, which are transferred to torque and bending moments by an anisotropic layer. **c** Design of magnetically guided remote shape-sensing microrobots. **d** Conceptual diagram showing the mechanism of the remote temperature-stimulated shape-sensing microrobot. Rotating magnetic fields navigate the microrobot. A thermoresponsive hydrogel layer shrinks at a phase change temperature (i.e., 40 °C), causing the helical microrobot to reconfigure its shape to planar. This change in antenna shape induces a switch in RF signal responses, enabling remote shape detection.

IONP/pNIPAM resin is added to fill the remaining gaps, while magnetic fields are applied to align the IONPs. After curing the hydrogel patterns with UV light, the layered structures are lifted off by removing both sacrificial layers (Fig. 2d). This microfabrication process demonstrates straightforward scalability, as it relies entirely on successive photolithography-based techniques (Fig. 2e). Furthermore, this integration concept is flexible and can be extended to accommodate other electronic components (Supplementary Fig. 3) or support alternative layer configurations.

Following this fabrication protocol, we designed the microrobot's layered structure based on classical lamination theory (detailed angular design parameters are provided in Supplementary Fig. 4). The base layer consists of the electronics layer (thickness 100 nm, pattern orientation: 90°), followed by a 10 μm-thick supporting layer, and a 10 μm-thick anisotropic scaffold layer with +45° orientation. The top layer (up to 20 μm thick) is composed of a pNIPAM hydrogel composite with embedded IONPs aligned at −45°. While each layer has its own preferred directionality, the anisotropic scaffold layer induces the formation of a helical structure upon hydrogel swelling, as predicted by classical lamination theory. As a result, when the laminated layered structures swell, they form a helix (Fig. 2f, g), regardless of the orientation of the electronics or IONP/hydrogel layers. The microscopy images (Fig. 2h, i) reveal that the helical axis tends to align closely with the scaffold axis. Adjusting the IONP concentration in the hydrogel layer only slightly alters the angle between the helical axis and the scaffold axis, from 4.7° to 10.3° (Supplementary Fig. 5). This indicates that the scaffold layer plays a dominant role in determining the helical shape, fully transferring any shape change of the soft robotic structure to the dipole antenna. This helical design also ensures that, after shape morphing, the IONPs are vertically aligned with the helical axis, which is critical for generating corkscrew motion under a rotating magnetic field.

## Design parameter study for effective magnetic navigation and shape reconfiguration

The formation of helical structures under swelling is additionally governed by the length and width dimensions of the planar form. To confirm which sizes were appropriate, we varied the dimensions between 1 and 2 mm of width, and 5, 10, and 15 mm lengthwise. As seen in Fig. 3a, planar structures of either 1 mm or 2 mm in width and at least 10 mm in length formed helical structures under swelling. Thus, 2 mm × 15 mm and 2 mm × 10 mm are adequate designs for magnetic navigation through corkscrew locomotion, which agrees with previous research showing that higher length-to-diameter ratio helical microrobots have an advantage in generating corkscrew locomotion under rotating magnetic fields[54]. Additionally, while longer microrobots of the same width resulted in more turns in the helical shape, wider microrobots of the same length exhibited a lower helical angle due to reduced bending moment and torque. As such, one should note that the aspect ratio (ratio length to width of the planar structure) plays an important role in converting volumetric change to bending moment and torque. The 2 mm × 15 mm microrobot showed the highest length-to-diameter ratio (~4.1) compared to others, whereas the 2 mm × 10 mm microrobot follows with a ratio of ~3.5.

We navigated the 2 mm × 15 mm microrobot along a zigzag-shaped water channel under a rotating magnetic field (40 mT at 4 Hz, Supplementary Movie S3), with an average speed of 0.6 mm s⁻¹, using a magnetic navigation system consisting of three pairs of Helmholtz coils (details in the "Methods" section and Supplementary Fig. 6). Note that microrobots can also be manipulated in dragging motion using gradient fields, which can particularly be useful for controlling planar shape without relying on corkscrew motion capability (see Supplementary Fig. 7 and Movie S4). The average speed of 2 mm × 15 mm microrobot was 1.2 mm s⁻¹ under a magnetic field of 10 mT and gradient of 120 mT mm⁻¹.

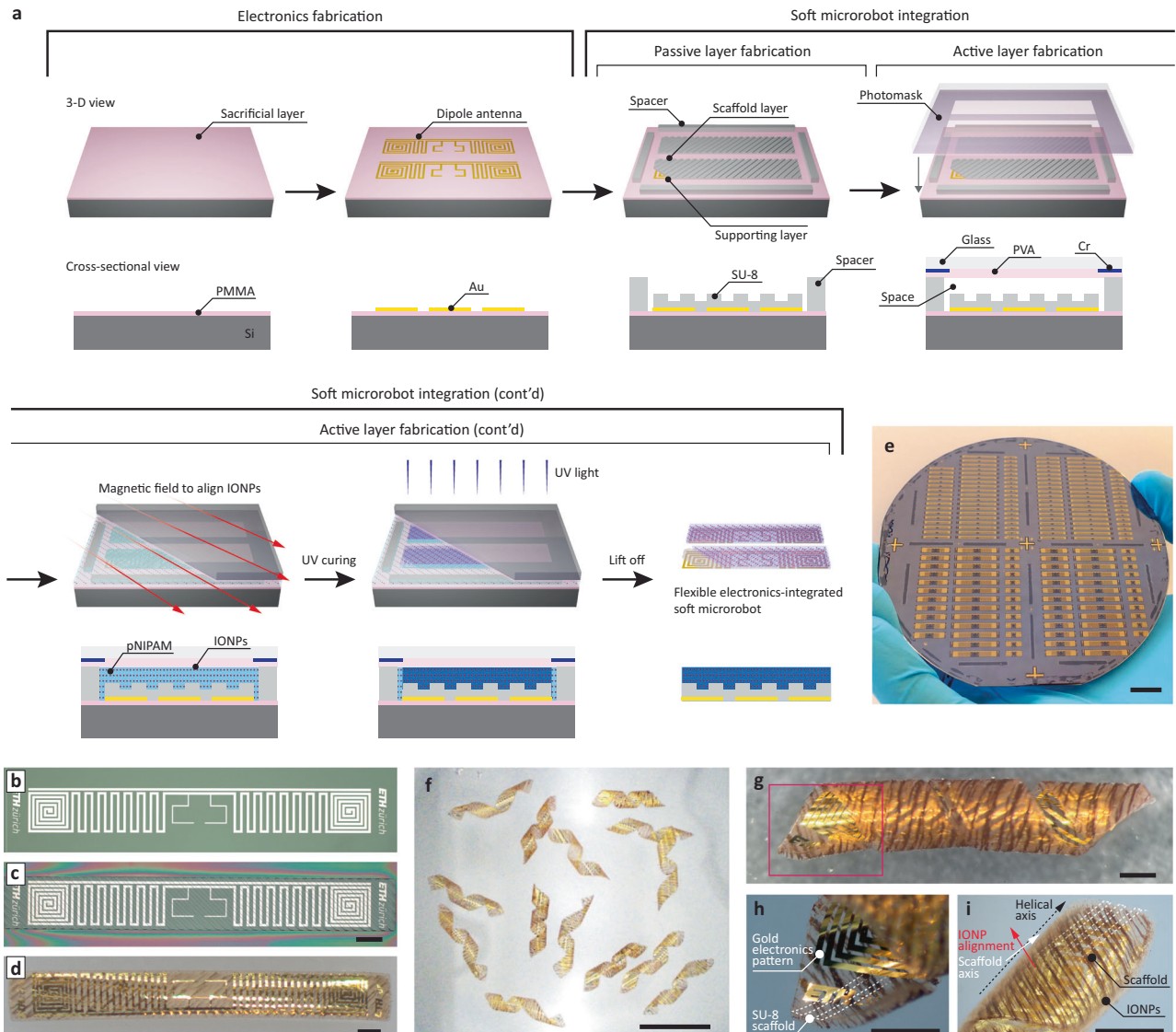

**Fig. 2 | Integration of flexible electronics into soft microrobots.**
**a** Microfabrication process to integrate three types of layers. After dipole antenna fabrication onto a sacrificial layer, passive and active layers are fabricated by photolithography processes. By removing the sacrificial layer using organic solvent, the integrated microrobots lift-off. **b** Optical image of electronics fabrication result. **c** Optical image of passive layer fabrication result (Scale bar: 1 mm). **d** Optical image of lift-off result (Scale bar: 1 mm). **e** Optical image of integrated microrobots, showing over 180 units on a wafer (Scale bar: 10 mm). **f** Optical image of fabricated helical microrobots (Scale bar: 10 mm). **g** Magnified image of a helical microrobot. The IONPs are aligned vertical to the helical axis, creating corkscrew locomotion through rotating magnetic fields. **h, i** Magnified optical images of the highlighted as red box in (**g**). Scale bars in (**g–i**) indicate 1 mm.

We also confirmed the shape reconfiguration capability, which switches the shape from helical to planar in response to temperature change. A gradual increase in temperature to 40 °C resulted in a flattening of the shape across all designs (Supplementary Fig. 8), which was repeatable over several attempts (Supplementary Fig. 9 and Movie S5). Additionally, the repeatability of shape reconfiguration was evaluated by analyzing the average body length and its standard deviation. We found that the average body length of planar microrobots varied from 14.9 mm to 14.4 mm over two rounds of shape morphing, while that of helical microrobots ranged from 11.4 mm to 10.9 mm. When the average length of the planar microrobots is divided by the designed body length, it results in an error of -0.6-4%, indicating that shape sensing can be repeatable. Furthermore, the consistency of shape morphing among multiple microrobots was assessed using the standard deviation-to-average ratio (i.e., coefficient of variation) of body length across three microrobot units. The planar shape shows coefficient of variation values of -0.038 and -0.063 over two rounds of

shape morphing, indicating a stable shape reconfiguration across the microrobots in response to environmental temperature changes.

## Shape detection performance through RF communication
We evaluated the shape detection performance through RF communication for the above microrobot designs. The design discussed above is intended to exhibit significant signal shifts based on the geometry state (helical or planar). The scattering transmission coefficient from port 2 to port 1 (i.e., $S_{21}$ signal) was acquired using a vector network analyzer (VNA) under an experimental configuration described in Fig. 3b (details in the "Methods" section and Supplementary Fig. 10). The gap between coils was set to 10 cm. Figure 3c shows the comparative RF signal shifts upon shape change. In the study, microrobots with dimensions of 1 mm × 15 mm, 2 mm × 10 mm, and 2 mm × 15 mm demonstrated significant signal differences between helical and planar shapes as shown in Fig. 3c. Additionally, we observed that the microrobots had a RF resonance frequency near 12 GHz in air. Based on

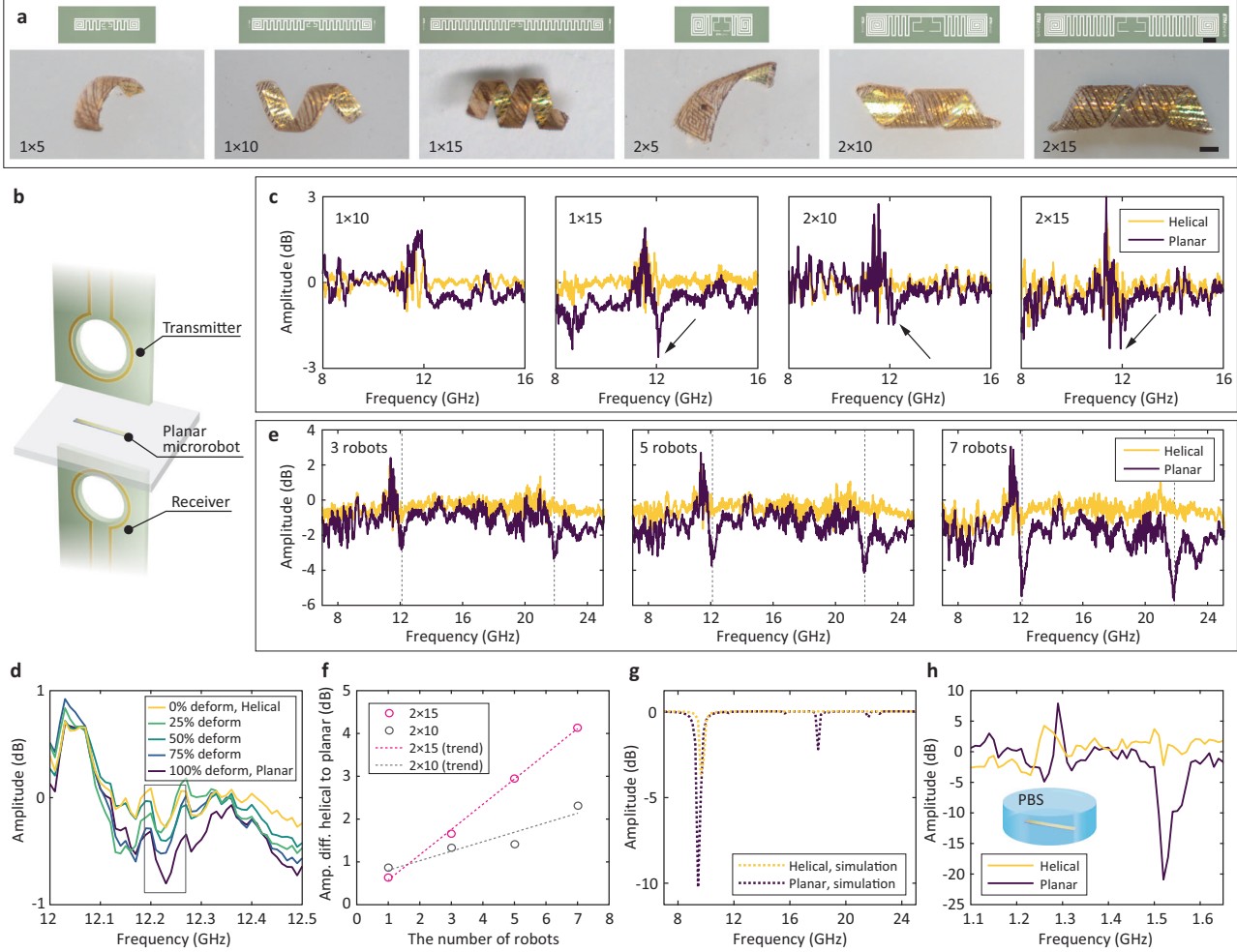

**Fig. 3 | Design parameter study and shape-detecting performance through RF communication. a** Helical shape formation varied by the design parameters (e.g., 1 × 5 indicates a 1 mm × 5 mm sized microrobot.). Scale bar indicates 1 mm. **b** Schematic diagram of experimental setup for RF communication performance evaluation. **c** RF signal shifts according to shape reconfiguration from helical to planar. 1 × 15, 2 × 10, and 2 × 15 microrobots show both signal shift and resonance. **d** Real-time RF signal monitoring during continuous shape deformation from helical to planar, showing signal amplitude increase throughout the transition.

**e** Collective effect on RF signal change with 2 × 15 microrobots. As the number of microrobots increased, the resonance was clearly observed, implying that an increase in the number helps determine the shape through RF communication. **f** Comparison of collective effect between 2 × 10 and 2 × 15. **g** Simulation results of RF signals of helical and planar microrobots. **h** RF signal shifts according to shape reconfiguration from helical to planar of seven units of 2 × 15 microrobots immersed in PBS solution. The resonant frequency of the microrobots in PBS solution shifts to ~1.5 GHz, whereas ~12 GHz in air.

this observation, we took advantage of the resonance peak to distinguish the shape of microrobots over barriers by focusing our analysis around 12 GHz. Considering both shape reconfigurability and shape detection performance, 2 mm × 15 mm and 2 mm × 10 mm microrobots were selected for further evaluation.

Microrobots with greater area exhibited more distinct signal differences between helical and planar states. This phenomenon can be analytically validated to support the experimental results and to further refine antenna designs. Assuming an unobstructed line-of-sight propagation in free space, the Friis transmission equation describes the power transfer relationship between two antennas (i.e., transmitter and receiver coils) as:

$$\frac{P_r}{P_t} = G_t G_r \left(\frac{\lambda}{4\pi R}\right)^2 \tag{1}$$

where $P_r$ and $P_t$ are the power received by the receiver and the power transmitted from the transmitter, respectively. $G_t$ and $G_r$ are the gains of the transmitter and receiver, respectively. $\lambda$ is the wavelength, and $R$

is the distance between the transmitter and receiver[55]. When the dipole antenna is placed between the transmitter and receiver coils, transmission in radio communication involves a two-step procedure, and the transmission coefficient ($S_{21}$) is defined as:

$$|S_{21}|^2 = \frac{P_r}{P_t} = \eta^2 G_t G_r \left(\frac{\lambda}{4\pi R_1}\right)^2 \left(\frac{\lambda}{4\pi R_2}\right)^2 D_0^{\ 2} \tag{2}$$

where $\eta$, $R_1$, $R_2$, and $D_0$ are the radiation efficiency of the dipole antenna, the distance between the transmitter and dipole antenna, the distance between the dipole antenna and receiver, and the maximum directivity of the dipole antenna, respectively. The maximum directivity ($D_0$) is defined as:

$$D_0 = \frac{4\pi}{\lambda^2} A_{em} \tag{3}$$

where $A_{em}$ is the effective aperture of the antenna. Consequently, the shape transformation of microrobots from a helical to a planar shape

increases the effective aperture ($A_{em}$), enhancing the maximum directivity ($D_0$) and shifting the transmission coefficient ($S_{21}$) curve (details for theoretical estimation in Supplementary Methods). Finally, these signal changes additionally scale with the absolute physical area of the microrobot. This was demonstrated by continuously tracking the RF signals as a single microrobot's shape morphed from helical to planar (shown in Fig. 3d). During the shape transition, the amplitude at the resonance frequency gradually shifted by an order of $10^{-1}$, highlighting the microrobot's capability for real-time shape sensing.

To make sure that the signal effects were primarily from the antenna, we experimentally confirmed that the IONPs had a negligible effect on RF signal change. Standalone electronics (i.e., antenna onto the SU-8 supporting layer) and the electronics integrated with microrobots (i.e., antenna with the SU-8 supporting layer and IONP/hydrogel composite layer) showed similar signals (Supplementary Fig. 11a), which indicates that the influence of IONPs on radio communication can be disregarded when designing microrobots.

## Enhanced shape detection performance through collective effect

We investigated whether the collective effect of many microrobots could improve the shape detection performance through RF communication, using the 2 mm × 15 mm sized microrobots (Fig. 3e and Supplementary Fig. 12a). Increasing the number of microrobots led to a greater distinctive difference in RF signal amplitude between helical and planar shapes at ~12.02 GHz and ~21.70 GHz. The maximum amplitude difference increased from 1.3 dB at 11.97 GHz to 4.1 dB at 12.07 GHz (near the first RF signal peak) and from 1.7 dB at 21.60 GHz to 5.2 dB at 21.80 GHz (near the second RF signal peak). Figure 3f shows the signal changes as a function of the number of the microrobots at 12.06 GHz. Additionally, similar results were observed with 2 mm × 10 mm sized microrobots (Supplementary Fig. 12b). The slope of trend lines, indicating sensitivity enhanced by the collective effect, further suggests that the 2 mm × 15 mm microrobots could provide improved performance in remote shape detecting through collective microrobots.

The collective effect of multiple dipole antennas on transmission performance is also theoretically supported. The maximum directivity $D_0$ of an array of antenna components with small spacing ($d \ll \lambda$) is defined as:

$$D_0 \approx \frac{Nkd}{\pi} = 2N\left(\frac{d}{\lambda}\right) \quad (4)$$

where $N$, $k$, and $d$ are the number of antenna components, Boltzmann constant, and space between components, respectively. As the number of components ($N$) increases, the maximum directivity ($D_0$) increases, enhancing the transmission coefficient $S_{21}$ from Eq. (2).

Additionally, we performed a full wave 3-D finite element method simulation to further confirm that multiple antennas' shape transformation can induce the RF signal changes using COMSOL's Multiphysics RF Module. Figure 3g shows a comparison of simulated RF responses of planar- and helical-shaped dipole antennas. The comparison with experimental data (Fig. 3e) reveals a similar trend in resonance frequency with an offset, confirming that experimental RF signal evaluation results are consistent with the analytical results.

In addition to the increase in RF signal differentiation, we also verified two important aspects of how the signal behavior changes from collective effects. First, the RF response from standalone dipole antennas were consistent with those observed from the integrated microrobotic systems (Supplementary Fig. 11b, c), which implies that the dipole antenna effects were dominant over any contribution from the IONPs. Secondly, we also took into consideration the role transmitter distance could play. Most notably, we observed that the distance between the transmitter and receiver is non-linearly related to the radio communication performance in collective microrobot signaling. For collections of 7 microrobots, we got better performance out of a receiver placed at 10 cm compared to 5 cm. Whereas for 5 or fewer microrobots, the 5 cm distance was more optimal (Supplementary Fig. 12c, d). This result could be explained by the far-field properties of dipole antennas and suggests that receiver distance should be optimized based on the number of microrobots for better shape detection performance.

## Investigation of environmental effects

The performance of RF communication-based shape detection can be significantly influenced by environmental factors, including the medium (e.g., ion-containing aqueous solutions) and the surroundings (e.g., experimental setups). First, different media can shift the resonant frequency of RF signals. The basic mechanism of the dipole antenna's resonance is based on the matched wavelength of the electromagnetic wave. The wavelength at the resonant frequency is defined as:

$$\lambda = \frac{c}{f_{r,\,air}} \quad (5)$$

where $c$ and $f_{r,\,air}$ are the speed of light and the resonant frequency in air, respectively. In a specific medium, the speed of light $v_m$ is given by:

$$v_m = \frac{c}{n} \quad (6)$$

where $n$ is the refractive index of the medium. Based on Maxwell's electromagnetic field theory, the refractive index is given by $n = \sqrt{\varepsilon_r \mu_r}$. For non-magnetic materials, the relative magnetic permeability index $\mu_r$ is ~1, meaning that the refractive index depends solely on the relative dielectric constant $\varepsilon_r$. Thus, the refractive index simplifies to $n = \sqrt{\varepsilon_r}$. Since the electromagnetic wavelength for the resonance of dipole antenna needs to remain the same across different media (e.g., air and ion-containing aqueous solutions), Eqs. (5) and (6) can be rewritten as:

$$\frac{c}{f_{r,\,air}} = \frac{v_m}{f_{r,\,medium}} = \frac{c/n}{f_{r,\,medium}} \quad (7)$$

where $f_{r,\,medium}$ is the resonant frequency of the dipole antenna in a specific medium. Thus, the resonant frequency $f_{r,\,medium}$ is defined by Eq. (8).

$$f_{r,\,medium} = \frac{f_{r,\,air}}{\sqrt{\varepsilon_r}} \quad (8)$$

Considering the developed 2 mm × 15 mm microrobots, they exhibited a resonant frequency of ~12 GHz in air. When immersed them in phosphate-buffered saline (PBS, relative permittivity: ~70), the resonant frequency is expected to shift to ~1.45 GHz. Figure 3h shows that experimentally measured resonant frequency shifts to ~1.52 GHz in the PBS medium, confirming the above analytical estimation is reasonable. This analytic estimation and experimental results also aligned with a simulation result, which shows a resonant frequency shifts to ~1.47 GHz (Supplementary Fig. 13).

We also examined the effect of the experimental setup as a method to investigate surrounding influences. In the setup, the sample holder (i.e., a plate to place microrobots in the experimental setup) can affect the measured RF signals, as it is positioned between the antennas and can modulate signals. Note that we excluded effects of antennas, VNA, and cables, as they are pre-calibrated, making their contributions to signal variability negligible. First, we compared RF signals in two configurations of the experimental setup, that is, with and without the sample holder. As a result, overall trends of RF signals were similar across both experiments, while slight amplitude shift was

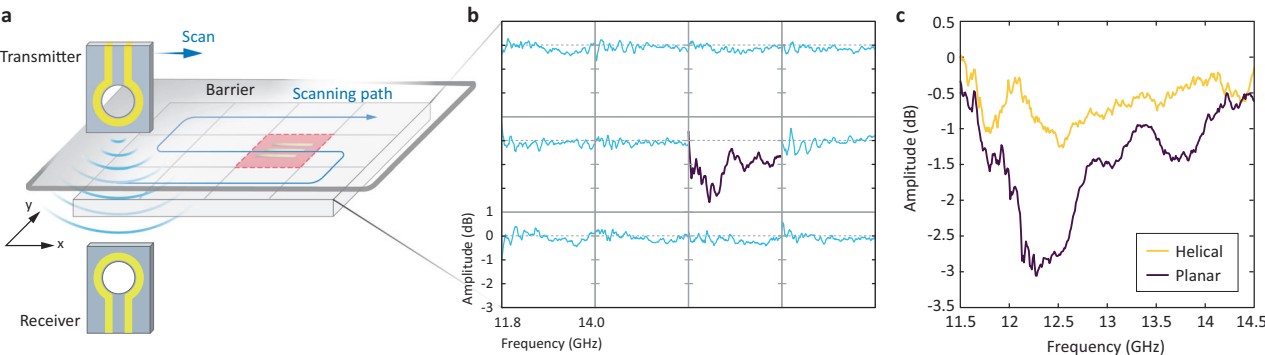

**Fig. 4 | RF communication-based microrobot localization. a** Schematic of RF communication-based localization, consisting of a scanning system and a sample holder divided into four-by-three array. The localization system consists of transmitter and receiver coils positioned opposite to each other, with scanning performed along a predefined path. **b** RF signals across the array exhibit a distinctive signal at position (3,2) corresponding to the location of the planar microrobots. **c** RF signals from seven units of 2 × 15 microrobots placed behind a barrier. The clearly distinct signals between helical and planar shapes demonstrate that the microrobots can be detected through the barrier.

observed (Supplementary Fig. 14a). Additionally, RF signals from different sample holder designs also show negligible effects (Supplementary Fig. 14b), demonstrating consistent shape detection performance across different experimental setup configurations. These findings indicate that while the sample holder may introduce minor amplitude deviations, it does not cause additional signal artifacts (e.g., extra peaks or dips). Any small shifts caused by the setup can be readily corrected during data processing.

### Microrobot localization

Wireless communication capability enables microrobot localization and local information perception. Figure 4a illustrates that planar microrobots can be localized through areal scanning using transmitter and receiver antennas. This antenna setup scanned a sample holder divided into a four-by-three array and detected a notable dip (at ~12.2 GHz) in the measured RF signal at coordinate position (3,2), where the planar microrobots were positioned (Fig. 4b; see also Supplementary Fig. 15a).

Additionally, when the microrobots were placed under a cover (Supplementary Fig. 15b, c), the planar and helical microrobots transmitted distinguishable RF signals, enabling shape detection (Fig. 4c). This demonstrates that planar microrobots can be detected via RF communication without the need for a vision system. Finally, these demonstrations imply that developed microrobots can be localized even when covered, allowing their use for a local shape sensing.

### Demonstration of microrobots' actuation and remote shape recognition

To demonstrate the complete operation of the proposed microrobot system, we constructed a functional example of the process (Fig. 5 and Supplementary Movie S6). We positioned three helical microrobots (2 mm × 15 mm) in a water channel with varying temperatures, simulating magnetic navigation, thermoresponsive shape reconfiguration, and RF-based shape recognition (Fig. 5a). The microrobots were actuated under rotating magnetic fields (40 mT at 4 Hz) using a magnetic navigation system (experimental configuration in Supplementary Fig. 6), propelled along the water channel from the low-temperature zone to the high-temperature zone via corkscrew motion (Fig. 5b). Upon reaching the high-temperature zone (~40 °C), the microrobots transformed from a helical to a planar shape (Fig. 5c).

The shape reconfiguration from helical to planar was evaluated as a function of temperature (Fig. 5d). These reconfiguration results were then correlated with the RF signal amplitude at the resonant frequency. The phase change temperature was set to 40 °C, as the shape

reconfiguration from helical to fully planar was completed within a reasonable timeframe (less than 10 s) under this condition (Supplementary Fig. 16). Based on this analysis, we remotely detected the microrobot's shape states (Fig. 5e, f) through their RF responses using the VNA. By comparing the measured RF responses (Fig. 5g) with our reference data set, we confirmed the successful shape reconfiguration of the microrobots. Notably, the RF response corresponding to the fully planar shape (pink line in Fig. 5g) reflects deformation triggered by elevated temperatures (~39–41 °C), indicating that passive, binary inference of environmental temperature may be feasible based on the final microrobot shape. Note that experiments were performed sequentially for i) navigation and transformation and then ii) transformation and temperature sensing, as the Joule heating coils in the navigation setup can distort the radio communication signals.

## Discussion

We demonstrated remote communication with soft, shape-reconfigurable microrobots, featuring a key innovation in the integration of a flexible dipole antenna. The microrobot's environment-responsive shape-changing ability directly translates into antenna reconfiguration, providing a clear mechanism for detecting its morphological state remotely. By designing the antenna to match the microrobot's dimensions, we also significantly enhance the sensitivity and reliability of radio communication signals. The integration of flexible electronics with smart materials addresses the signal limitations commonly observed in previous micro-antenna systems. As a result, this microrobotic platform functions simultaneously as an actuator and transducer, eliminating the need for strain gauges or wired power sources.

In our manufacturing process, we enhanced the spacer-method-based microfabrication method previously developed by our group[8] to integrate a flexible dipole antenna into soft microrobotic structures. Specifically, functionalizing the SU-8 surface through techniques like bond cleavage and silanization was crucial for achieving complete shape transformation between the planar and helical states. Since our manufacturing protocols are based on the well-established microfabrication processes, they are compatible with different semiconductor technologies[56], including MEMS[57,58] and flexible electronics[59,60] processes by combining the electronic elements prior to fabricating the microrobotic structures. This means other electronic devices (e.g., thin-film transistors[61,62], sensors[63], electroluminescent[64], and energy harvester[65]) can be used to expand the microrobot's functionality and take advantage of even higher levels of intelligence (e.g., processing, computing). Also, the microfabrication processes are

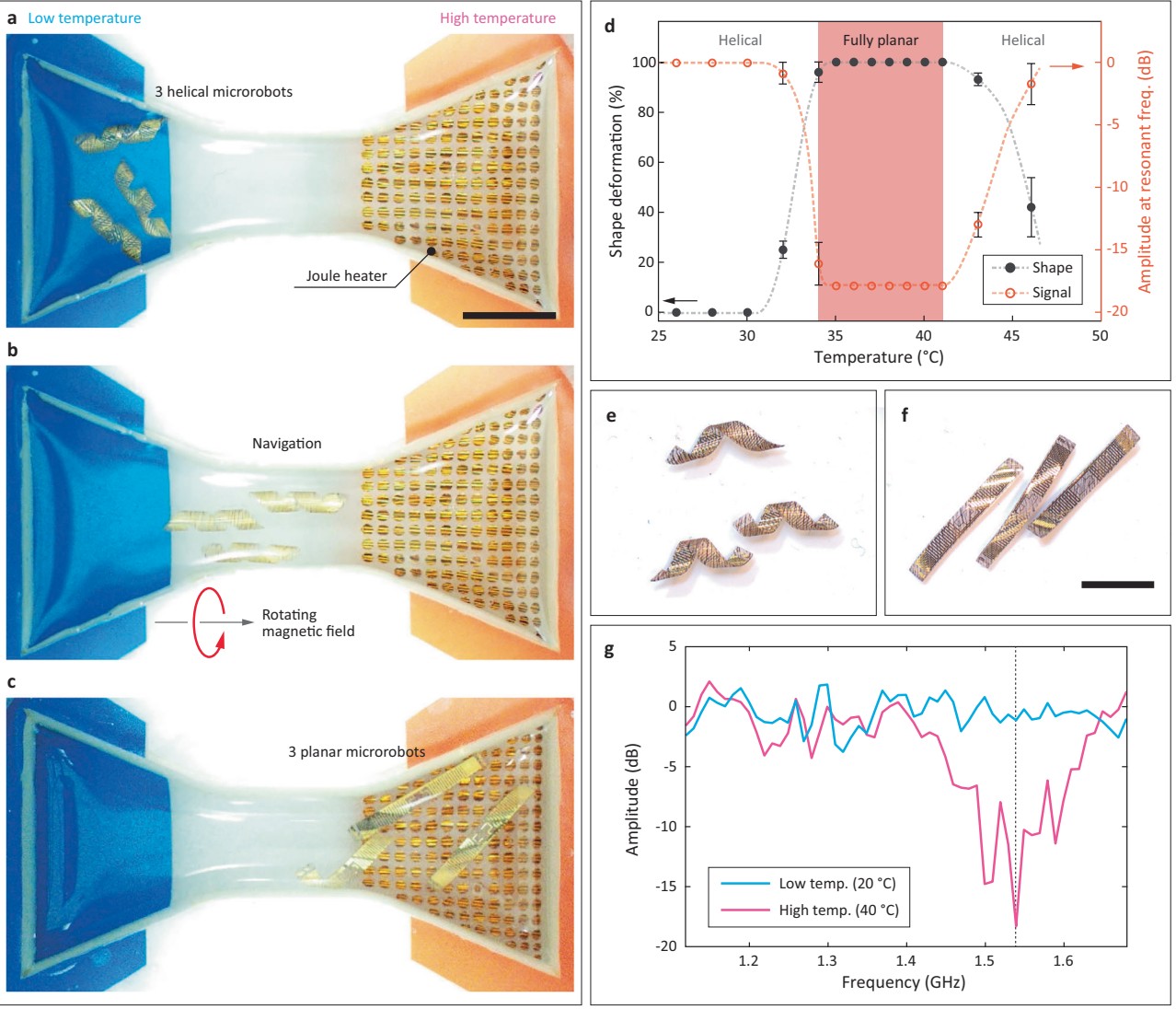

**Fig. 5 | Demonstration of magnetic navigation, thermoresponsive shape reconfiguration, and remote shape-sensing with three microrobots. a, b** The rotating magnetic field navigates microrobots from a low-temperature zone to a high-temperature zone. **c** The helical microrobots deform to planar microrobots at the high-temperature. **d** Shape deformation from helical (0%) to planar (100%) as a function of PBS temperature. The signal amplitude at resonant frequency is estimated from the shape. Error bars represent standard deviation. **e, f** Optical images of fully helical and fully planar microrobots under the RF communication performance evaluation setup. **g** Shift of RF signal determines that environmental temperature is ranged 39–41 °C. Scale bars indicate 10 mm.

scalable and suitable for mass manufacturing with batch-processing options, producing microrobots of varying sizes and designs.

Additionally, we explored various pathways for improving the functionality of the developed microrobots. For example, different antenna designs produced measurably shifted resonant frequencies. Signals from two types of microrobots (2 mm × 15 mm and 2 mm × 10 mm with a 10 cm gap between the transmitter and receiver antennas) displayed a difference in resonant frequency shifts of ~300 MHz (Supplementary Fig. 12a(iv) and 12b(iv)). Furthermore, $S_{21}$ varied with different antenna designs. As shown in Supplementary Fig. 17, the square spiral antenna exhibited a resonant frequency of ~20.6 GHz. Taken together, it becomes clear that careful antenna design can be used as a tool to distinguish microrobots, which when used in denser distributions, could enhance the signal sensitivity, as shown in Supplementary Fig. 18.

We also theoretically and experimentally confirmed that the use of microrobots in the ion-containing aqueous solution medium significantly lowered the resonant frequency and enhanced the RF communication performance, where one-order reduction in resonant

frequency enhances the penetration capability of RF signals. Comparing the RF signals of the dipole antenna in PBS at 2 mm and 4 mm depths shows a clearer signal distinction between signals at a greater depth (Supplementary Fig. 19). This is probably because the increased medium thickness absorbs more RF signals[66], enhancing signal sensitivity. Since human body fluids are similarly ion-containing aqueous solutions, there is potential applicability for these microrobots in biomedical applications. Additionally, tests simulating microrobot movement within a two-hop relay architecture (i.e., an antenna-embedded microrobot positioned between fixed transmitter and receiver coils) showed that $S_{21}$ variation due to position was negligible compared to that caused by shape reconfiguration, suggesting potential applicability even for movement along the depth axis (see Supplementary methods and Supplementary Fig. 20). However, their effectiveness in biological settings remains to be evaluated in future studies.

Continuous tracking of the microrobot's shape demonstrates the feasibility of this approach for real-time, remote monitoring of the microrobot's morphological state. This enables remote recognition of

the microrobot's configuration (e.g., helical vs. planar or folded vs. unfolded) during operation, addressing one of the key challenges in the practical use of microrobots. For example, configuration recognition can be advantageous in targeted delivery applications, where active shape reconfiguration (e.g., via magnetic hyperthermia induced by alternating magnetic fields) is required to adapt to complex environments. In such cases, remote communication can inform successful shape transformation, allowing the microrobot to proceed to its next task. Additionally, our microrobot localization experiments demonstrate that the system can reliably detect both position and shape, suggesting its potential for local temperature sensing applications.

The hydrogels used in this study swell at the 34–37 °C of lower critical solution temperature (LCST) with a narrow LCST range[67], which can be further tuned by copolymerizing hydrophilic and hydrophobic monomers and/or by modifying the polymer architecture. By manipulating the swelling behavior, the detectable temperature range can be varied[68]. Based on this, the temperature sensing accuracy in the demonstration, which is originally limited by the intrinsic nature of hydrogel-based temperature sensing, may be further improved through a fuzzy logic approach, leveraging the collective effect of microrobots designed with varying phase change temperatures. Additionally, the type of hydrogels can be extended to various stimulus-responsive hydrogels (e.g., pH, light, electric, chemical) beyond the heat-detection demonstrated here. Thus, this microrobotic concept has the potential to be expanded into a wide range of devices for monitoring diverse stimuli at the microscale.

Taking the remotely controlled actuation and local environmental sensing together, one begins to see an interesting use case for the developed microrobotic platform in biomedical applications (e.g., binary temperature sensing in localized diagnostic applications). Regions of tumor or inflammation exhibit higher temperatures, typically elevated by 0.5–3 °C. As an initial step, we examined the feasibility of using this microrobot in bio-settings by testing the material biocompatibility with MTT assays (Supplementary Fig. 21). When exposed to human umbilical vein endothelial cell (HUVEC) cultures for 24 and 72 h, the microrobots demonstrated over 80% cell viability, indicating acceptable levels of biocompatibility. Meanwhile, studying radio communication in bio-settings reveals both promising aspects and ongoing challenges. For example, the penetration depth of RF electromagnetic waves (1–5 GHz) through tissues falls within a reasonable range (16–60 mm). However, signal reflection at the air-tissue interface remains a limitation, indicating the need for further optimization of external transmitter and receiver coil designs. In addition, while increasing transmitting coil power can help mitigate signal attenuation over these depths, human exposure to RF electromagnetic fields should be considered. Although research is ongoing, power levels can be increased within safe limits defined by the tissue heating capacity, measured by the specific absorption rate (SAR), which is generally accepted to be less than 2 W kg$^{-1}$ for localized exposure[69]. Additionally, it is known that the RF ablation power levels are in the watt range and are used continuously for several seconds[70]. Currently, our system operates at ~1 mW for communication, which is 10$^4$ times lower than the power typically used for ablation, indicating that there is room for safely increasing the transmission power if needed. According to the IEEE standard for safety levels with respect to human exposure to electromagnetic fields, the local exposure reference level at 10 GHz is ~182 mW cm$^{-2}$, which is substantially higher than the levels used in this study[71]. Furthermore, in the working scenario, perception and communication can operate with a 10% duty cycle (e.g., 10 ms on and 90 ms off), which can also manage RF heating. Combined, this wireless communication-capable microrobotic platform holds promise for future medical use.

Our device stands out from existing sensing and communication methods in several key ways. First, by leveraging flexible electronics, the proposed microrobotic system offers significant advantages over conventional rigid CMOS and RFID devices, including flexibility, adaptability, and enhanced safety and compatibility in diverse environments. While acoustic sensing and communication present a potential alternative[72,73], they tend to capture information over a broad area, making them less suitable for localized perception. Additionally, existing acoustic communication schemes often rely on rigid implantable structures and are primarily focused on localization, without the ability to externally transmit localized sensory data.

In summary, our proposed microrobot design and manufacturing protocol, which integrates flexible electronics with soft microrobots, enables the development of fully flexible, wireless-sensing microrobots capable of magnetic navigation and collective behavior. The nanometric thin dipole antenna remains fully functional under the mechanical strain induced by shape transformation of the microrobots, transmitting different radio communication signals between planar and helical shapes. The collective behavior further enhances the signal differentiation, ensuring reliability in RF-based shape sensing. We expect that this advancement could show possibilities in embodied micro-intelligence that can interact with the environment and gather data, thereby facilitating the autonomous operation of small-scale robots.

## Methods

### Materials
SU-8 and Polymethyl methacrylate (PMMA) photoresists were purchased from Kayaku Advanced Materials, Inc. (USA). 30-nm diameter iron oxide nanoparticles (IONPs) coated with polyvinylpyrrolidone (PVP) were purchased from US Research Nanomaterials Inc. (USA). N-Isopropylacrylamide (NIPAM, 97%), N,N'-Methylenebis(Acrylamide) (BIS, 99%), 1-Butanol (99.9%), Poly(vinyl alcohol) (PVA, 80%, MW 9k), Propylene glycol monomethyl ether acetate (PGMEA, 99.5%), and Acrylic acid (AAc, 99%) were obtained from Sigma-Aldrich (USA). Poly-N-Isopropylacrylamide (pNIPAM, MW 300k) was purchased from Scientific Polymer Products Inc. (USA). The photoinitiator Omnirad 2022 was supplied by S u. K Hock GmbH (Germany). All chemicals were used as received.

### Preparation of magnetic thermoresponsive hydrogel
The magnetic thermoresponsive hydrogel was prepared by mixing NIPAM (monomer), BIS, 1-Butanol, and AAc in a molar ratio of 118:5:365:12. The photoinitiator was added as 1 wt% of the entire hydrogel matrix. The mixture was stirred overnight. Before use, IONPs were mixed into the hydrogel at a concentration of 7 wt%.

### Integration of flexible electronics and soft microrobot
A 4-inch silicon wafer was spin-coated with a 2 μm-thick PMMA acting as a sacrificial layer. A 100 nm-thick gold layer was deposited on the wafer using an electron beam deposition process, followed by photolithography, wet etching, and cleaning processes to fabricate antenna patterns (trace width 100 μm, trace gap 50 μm). Subsequently, two SU-8 layers (10 μm thickness each) were patterned sequentially to fabricate a supporting layer and scaffold layer using a mask aligner (MA/BA6 Gen4, SUSS). Then, a 3 μm-thick SU-8 layer was fabricated as a spacer to control the overall thickness of the microrobots. The SU-8 surface was functionalized by silanization and silicon oxide coating to enhance adhesion.

Afterward, a magnetic thermoresponsive hydrogel layer was fabricated through a magnetic field-assisted photolithography process. A photomask was prepared using a 4-inch glass wafer coated with chromium. The chromium layer was patterned with the hydrogel layer design and then coated with a 2 μm-thick PVA sacrificial layer on the patterned side. This photomask was placed on the spacer, aligning the markers (Supplementary Fig. 2) using a custom-built magnetic field-assisted photolithography device (Supplementary Fig. 22). The magnetic thermoresponsive hydrogel was then injected into the space,

specifically between the silicon wafer and the glass wafer. This entire sandwich structure was then placed into a static uniform magnetic field of 10 mT for 4 min to align the IONPs, followed by exposure to UV light (wavelength: 365 nm, intensity: 3 mW cm$^{-2}$) for 2 min while maintaining the magnetic field. After curing, this sandwich structure was sequentially placed into water and PGMEA to dissolve the sacrificial layers, releasing the integrated microrobots from the wafer. The detailed fabrication protocol flow chart is depicted in the Supplementary Fig. 23.

## Characterization of antenna fabrication stability

The stability of the antenna fabrication was characterized using a probe station to measure the resistance of the antenna. By correlating the resistance with different designs, we ensured the consistency of the electronic properties (Supplementary Fig. 24) and achieved a 100% yield in electronics fabrication.

## Evaluation of shape reconfiguration performance

The shape reconfiguration performance was evaluated by alternately placing the microrobots in 20 °C and 40 °C water. Optical images were captured using commercial CCDs (a2A3840–45ucPRO and acA1920-150uc, BASLER Inc., Germany) for image processing.

## Actuation of microrobots through rotating magnetic fields

Rotating magnetic fields were generated using three pairs of Helmholtz coils in three-dimensional space (Supplementary Fig. 6). Magnetic fields were controlled by adjusting their strength (< 46 mT), rotational frequency (< 10 Hz), and direction.

## Evaluation of shape detection performance by RF communication

RF communication was performed using a vector network analyzer (VNA, model N5247B, Keysight, USA) with a measuring frequency range of 10 MHz to 26 GHz. A pair of transmitting and receiving coils (diameter: 7 mm) were placed parallel to each other with a gap distance of 5 to 10 cm and were coaxially connected to the VNA. Samples were placed on an acrylic plate between the coils.

The VNA measured two-port scattering parameters, where $S_{21}$ denotes the transmission coefficient from one port to the other. The $S_{21}$ parameters were obtained under various experimental configurations, including measurements without microrobots (background signal), with helical microrobots, and with planar microrobots. We evaluated the signal change corresponding to the shape change by subtracting the signals with microrobot from the background signals. The collective effect was also evaluated through the same configuration.

## Radio communication simulation

The simulation was conducted using COMSOL Multiphysics RF Module with a full wave 3-D finite element method. The helical- or planar-shaped dipole antenna was respectively placed in a bounding box with periodic boundary conditions to mimic multiple antennas in close proximity. Linearly polarized plane waves with frequencies ranging from 5 to 30 GHz were excited above the microrobot surface, and transmission of the electromagnetic wave with the same polarization was monitored below the antennas for all frequency sampling points.

## Data availability

All the relevant data are available within the Article and Supplementary Information. Source data are provided with this paper.

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

## Acknowledgements

This work received financial support by the Swiss National Science Foundation under project No. 197017. We thank Yuqi Liu, Dr. Hua Wang, Dr. Juerg Leuthold, Dr. Lucio Pancaldi-Giubbini, and Dr. Mahmut Selman Sakar for their technical instrument support. We also thank the FIRST laboratory at ETH for their technical support and the Cleanroom Operations Team of the Binning and Rohrer Nanotechnology Center (BRNC) for their help and support, especially Ute Drechsler. Q.G. and M.K. thank Dr. Tianyun Huang, Dr. Semih Sevim, and Dr. Xiang-Zhong Chen for their helpful discussion.

## Author contributions

Q.G., M.K., G.C., B.J.N., and S.P. conceived the project. M.K., Q.G., and C.V. designed the experiments and developed the methodology. Q.G.

optimized the fabrication process. Q.G., D.V.A., X.Z., H.Y., C.E., D.C., and F.C. performed the experiments and investigations. Q.G. and M.K. conducted the formal analysis. X.Z. performed radio communication simulations. N.M. and M.M. participated in discussions. M.K. and Q.G. prepared the figures and videos. M.K. and Q.G. wrote the original draft, all authors interpreted the results, and E.Z., X.Z., B.J.N., and S.P. reviewed and edited the manuscript. M.K., B.J.N., and S.P. supervised this project.

## Competing interests

The authors declare no competing interests.
