## [Transparent Peer Review file · Nature Communications]

Soft Magnetic Microrobots with Remote Sensing and Communication Capabilities

Corresponding Author: Dr Minsoo Kim

Version 0:

Reviewer comments:

Reviewer #1

(Remarks to the Author)

The integration of actuation, sensing, and communication in one small-scale robot is challenging and critical for promoting the intelligence of small-scale robot. Current works tend to focus on the programmable deformation and designable actuation, the function of sensing and communication are less explored. In this paper, the authors integrate antennas, magnetic hydrogels, and anisotropic elements onto a micro-scale robot using photolithography techniques. The robot can be precisely propelled to the target position, respond to changes in external temperature and magnetic fields, and provide feedback on the perceived environmental temperature via radio frequency, which exhibit more capabilities and enhanced intelligence. Furthermore, the proposed manufacturing method for the robot is versatile and sensing modules can be extended from temperature to humidity, light, chemistry, and other areas, laying the solid foundation for future biomedical applications. However, in-depth research is needed when regarding the robot's actuation and retrieval after detection, and there are some problems required to be solved before the acceptance.

(1) In the introduction, the authors mentioned that small-scale robots have made progress in coding and multimodality. The magnetic programming, anisotropic programming, and multi-function integration should be noted, just as exhibited in Sci. Adv.8, eabn8932(2022) and Nat Commun 10, 4087 (2019).

(2) In the part of the communication experiment, it was noted that an experimental platform exists between the transmitter and the receiver in addition to the robot. Whether the experimental platform affect the experimental results?

(3) New experiment environment should be designed to make the control of robot form a closed loop for better reflecting the advantages of magnetic field actuation and the importance of communication when the robot cannot be directly observed

(4) The author proposed that different types of electronics may be involved in the manufacturing process to highlight the scalability of the manufacturing method. Will these different electronics have variable signal effects during communication? If there are different responses, does it mean that the communication process is able to recognize multiple robots with different designs? Corresponding experiment could be designed to demonstrate this point.

(5) In the final experiment, the robot is flattened due to the increase of temperature and the signal is captured by an external device, whether the flattened robot continue to respond to external magnetic field for subsequent recovery?

Reviewer #2

(Remarks to the Author)

This manuscript presents a magnetically guided soft microrobot with integrated communication and environmental sensing capabilities. A proof of concept was demonstrated using a remotely controlled temperature-sensing microrobot. There have been numerous studies employing NIPAM-polymer for transformation and temperature sensing, rendering this work lacking in novelty. Consequently, I am unable to recommend this paper for publication in Nature Communications. Furthermore, there are several issues within the manuscript that require attention and revision by the authors:

1. In the section reviewing related work, the manuscript mentions previous studies but fails to provide an in-depth comparative analysis of these works. A clearer discussion of how this study builds on or improves upon existing research is needed to emphasize its innovation and advantages.

2. The temperature within the human body remains relatively stable, making it unlikely that a soft robot with temperature sensing capabilities would be necessary for human applications. Additionally, the size of the robot appears to be too large for such purposes. This prompts the question of which specific environment would require a soft robot with temperature

sensing capabilities, further questioning its intended purpose. Additionally, it is unclear why magnetic control is necessary for this device.

3. The robot undergoes deformation in response to temperature stimuli. Is the degree of deformation consistent across trials? How much unexpected variation in deformation impacts the accuracy of the sensing signal?

4. Deformation is triggered when the temperature exceeds a certain threshold. With such conditions, how can the robot be deployed into a real system?

5. In complex environments, such as natural or living settings, how can the sensitivity and accuracy of the sensor signals be ensured?

6. While using multiple robots enhances the sensing signal, the impact of inter-robot distance and relative orientation on signal quality requires further evaluation.

7. The simplified experimental conditions used in this study may not fully capture the challenges and limitations that microrobots could encounter in real-world applications. A broader range of scenarios should be tested to assess these factors more comprehensively.

8. The manuscript is insufficient to verify the practical application of the robot. Although biomedical applications are mentioned, the challenges of operating in complex biological environments, as well as biocompatibility issues, are not sufficiently addressed. There is also a lack of discussion or data on potential health risks, such as immune responses or toxicity, that microrobots might pose in living organisms.

9. The manuscript does not propose specific strategies or plans for mass production of the microrobots or for their integration into existing systems or processes. Guidance on how to scale production and facilitate practical deployment would be beneficial.

Reviewer #4

(Remarks to the Author)

Reviewer Comments for Manuscript NCOMMS-24-77214-T: Soft Magnetic Microrobots with Remote Sensing and Communication Capabilities

This article reports on the fabrication and testing of a new class of mm scale lithographically fabricated robot which combines the functionalities of locomotion, temperature sensing, and data transmission via RF signals, in a single device platform. The key advance is the combination of a temperature sensitive bending magnetic hydrogel/SU8 bilayer with a gold antennae layer, to realize a trilayer device can sense and report information about temperature via deformation of the antennae by the hydrogel. The authors outline the concept of operation for the device, the fabrication protocol by which the devices are built, the effect that bending has on RF signal amplitude, how the effect increases with number of devices, and demonstrate that devices can locomote using external magnetic fields to regions of varying temperature. Overall, the biggest strength of this paper is taking up the task of integrating multiple functionalities into a single robot. The following changes would likely improve the quality of the manuscript:

1. Remote sensing generally denotes that one can infer the specific value of temperature from the action of the device. It would be a lot easier to judge this work if data were included that quantifies how the detected radio signal depends on environmental temperature. Specifically, we feel that somewhere in the main text there needs to be a graph that shows temperature vs the authors preferred measurement from the antennae (i.e. resonance amplitude, resonance frequency shift). All we could find was the statement that the hydrogel undergoes a phase transition around 35C, and the data in 4F where robots are stated to unfold at 40C. Depending on the width of this transition, these devices could report across biomedically relevant temperatures, or be completely unsuited as a body temperature of 40C constitutes a medical emergency. Some data about how temperature can be extracted and quantified based on the device behavior would dramatically improve this work. Better still would be a thoughtful discussion about device accuracy and precision, given this is a completely analog reporting system.

2. 1-10 mm is a large length scale for microelectronics, allowing for a range of off-the-shelf sensing and communication modalities. For instance, the chips used to control commercial RFID antennas are typically just mm in size (i.e., ignoring the antenna, the control circuit that packages data can be well under the size of this robot). Likewise, there are numerous publications demonstrating CMOS temperature sensors and chips well under 1 mm in size. This begs the question: why this schema? Why not use an integrated circuit to measure and modulate RF signal directly? Or, for that matter, why not use acoustic communication, such as in neural dust? These alternatives would have significant benefits in signal reliability and accuracy; the authors should explain the merits of their approach, as compared to equally large devices.

3. The paper could use more discussion of the role of the environment in communication. 10 GHz RF waves penetrate weakly into water and by extension tissue (see Fig 2, <https://arxiv.org/abs/1306.5709>). Typically, the penetration depth is on the order of a mm. This seems like a major problem: if the best-case signal change for temperature reporting is on the order of a 1-10dB, the same effect could be produced by the device (or transmitter) moving roughly 1 mm (i.e. ~1 body length) deeper in tissue. How would this get de-convolved in practice? Likewise, some discussion of the power needed to operate the device at a given depth, and whether they would be safe, would be helpful. A 1 mm penetration depth would imply that nearly 43 orders of magnitude in power would be lost to reach 10 cm into tissue. How could this be overcome, especially for biomedical applications?

4. The device design and fabrication protocol are nice, especially the integration of the magnetic material. This is one of the best parts of the paper, but the extensive discussion of the specifics should be relegated to the supplemental information, in favor of a briefer overview. Additionally, many statements claim standard fabrication techniques are key advances and

should be done away with entirely. For instance: “Key advancements included ensuring precise layer alignment”, i.e. using a mask aligner? “Improving adhesion between layers”, i.e. using industry-standard adhesion promoters? “Optimizing the anisotropic layer and IONP pattern orientation”, i.e. magnetizing in the correct direction? “We develop a versatile microfabrication process that incorporates flexible electronics onto a soft microrobotic chassis using combination of thin-film deposition, photolithography, and wet etching techniques”, the versatility of microfabrication is well established. Similar statements appear throughout the manuscript and should be removed.

5. The authors forgo what might be an illuminating discussion about the fact that devices lose motility in high temperature regimes. One could argue that this is a feature, and that the device could be configured to operate in a “seek and destroy” scheme, where they travel around an environment until they reach a region of desired temperature, then planarize and remain there indefinitely. Given the vast expertise of the authors, especially in biomedical applications of microrobots, we’d love to hear them expand on the nuanced ways in which the interweaving of sensing and locomotion may be beneficial.

6. The discussion section could be sharpened. For example: “Previous approaches using smaller antennas faced challenges with inefficient radio communication, unreliable remote sensor powering, and poor signal differentiation in environmental monitoring due to inert sensing mechanisms.” It is not demonstrated in this work that any of these problems were solved. “Our research highlights the microrobots’ potential for biomedical applications requiring operation within the body’s enclosed cavities or conduits.” Only cavities that are large enough to accommodate the size of one or several devices (a few cm as seen in figure 4), and shallow enough that the RF signal can penetrate at order 10 GHz. “This approach could also be extended to develop interactive microrobots that respond to both internal and external stimuli, gather data, communicate with each other, and autonomously manage their functions.” It’s not clear how most of these behaviors will be achieved with the presented platform. If these statements are true, they need clarification and expansion. If not, they should be removed.

Version 1:

Reviewer comments:

Reviewer #1

(Remarks to the Author)

The Authors have addressed all my concerns, I recommend the acceptance of this manuscript.

Reviewer #2

(Remarks to the Author)

Reviewer #3

(Remarks to the Author)

Reviewer #4

(Remarks to the Author)

Reviewer Response for Manuscript NCOMMS-24-77214-T: Soft Magnetic Microrobots with Remote Sensing and Communication Capabilities

We appreciate the author’s detailed responses to our comments. Unfortunately, their response and changes made to the manuscript have not resolved our concerns and in some cases created further questions.

Rebuttal to Comment 1 Response: Here the authors explained that the absence of more detailed temperature sensing data is due to the fundamental properties of the deformable hydrogel sensing modality, i.e. that it is primarily supposed to sense binary temperature change above or below a small, biologically relevant temperature window of 36-40 degrees C.

Nonetheless, the authors also provide a graph showing the change in the response of the antenna as a function of deformation %, but offer no quantification of the change as a function of temperature. This leaves lingering questions about one could reliably infer temperature based on these partial deformations, given the small change in the RF response and the possibility that unaccounted forces in the robot’s environment could impede the continuous deformation response to temperature needed to achieve differentiation between sensed values or manipulate the robot directly, thus obscuring the measurement. These points need to be addressed to lend credibility to the system’s usefulness.

Rebuttal to Comment 2 Response: The authors contend that despite the vast improvement in temperature sensing capabilities in existing mm scale CMOS architectures, such systems are “incompatible” in vivo because they are rigid, a problem which is remedied by their device flexibility. This is false, as there is an extensive body of work on flexible, biocompatible, electronics platforms which can sense and respond to temperature [D. Kim, et al. “Stretchable and Foldable Silicon Integrated Circuits”, Science 320-5675, 2008] [J. Rogers, et al. “Materials and Mechanics for Stretchable Electronics”,

Science 327-5973, 2010] [J. Viveneti. et al. "Flexible, foldable, actively multiplexed, high-density electrode array for mapping brain activity in vivo", Nature Neuroscience 14, 2011] [T. Kim, et al. "Injectable, Cellular-Scale Optoelectronics with Applications for Wireless Optogenetics", Science 340-6129, 2013][G. Hills, et al. "Modern microprocessor built from complementary carbon nanotube transistors", Nature 572, 2019]. One of the key findings shown by these platforms is that while the modulus of the materials used may be higher than the materials in the authors' devices, the bending stiffness can be matched by making the device layer sufficiently thin, allowing useful materials for electronics, such as metals and semiconductors, to be incorporated into flexible structures. As such, we maintain that the authors should explain why their electing to use a deformation-based temperature sensor is advantageous, when there exist alternatives with extremely high degrees of functionality.

Rebuttal to Comment 3 Response: We are pleased to see that the devices operate better in environments more similar to biological media, but are still not convinced that in vivo sensing at useful depths will be possible at safe radiation levels. It seems to us that even when assuming a penetration depth of 60 mm at the resonance frequency, the order -10 dB signal differentiation between the helical and planar states (.316 intensity ratio) is the same signal differentiation that would be achieved simply by moving the antenna 70 mm deeper into tissue (or a variety of other obstructions). Given that the robot is designed to move around its environment, potentially at varying depths to search for high temperature areas, it is not clear to us how temperature sensing information could be reliably inferred in the dissipative media of the body, given the demonstrated signal response. One could plausibly increase the transmitter power to achieve better signal differentiation at these depths, but as the authors note there is a hard cap on this workaround set by radiative heating of the media which is both medically dangerous and self-defeating as a temperature sensor, because the media could heat due to the RF signals. Similarly, one could argue that the robot could be tracked to determine its depth, but this undercuts the value of RF sensing (why not just image the shape directly at that point). We feel that these serious drawbacks to an RF communication schema must be addressed or refuted in a more substantial way in the manuscript.

Version 2:

Reviewer comments:

Reviewer #4

(Remarks to the Author)

Generally we feel that the authors have addressed our concerns with their added content. Although we're still unsure about the value of this system vs. other sensing modalities or its practicality in a real world setting, the authors now make factually correct claims and have improved their placement of their work in the broader context of research.

We give two final suggestion to improve the paper quality.

1) It would be to better clarify what's being presented in their new supplemental section on page 24: namely it is unclear if these experiments were performed in air or in water (or a tissue phantom). We recommend the authors spell out these experiments more clearly, especially since if they were done in air, the argument is not very compelling.

2) The power safety argument feels disingenuous. The typical IEEE limit for safe incident power density in RF in the GHz range (surface or whole body) is around 1mW/cm² (c.f. <https://ieeexplore.ieee.org/stamp/stamp.jsp?tp=&arnumber=8859679>). If you're emitting 1mW from a 120mm transmitter, it sure seems like you've over the safe exposure limit. Likewise, a comparison to tissue ablation is not reasonable, since in that case you're deliberately trying to produce a thermal effect. This claim should be tempered: state what you power use and put that in the context of the *standard* RF limits for sensing, measurement, and communication.

Responses to Reviewers' Comments

Reviewer 1:

General comments: The integration of actuation, sensing, and communication in one small-scale robot is challenging and critical for promoting the intelligence of small-scale robot. Current works tend to focus on the programmable deformation and designable actuation, the function of sensing and communication are less explored. In this paper, the authors integrate antennas, magnetic hydrogels, and anisotropic elements onto a micro-scale robot using photolithography techniques. The robot can be precisely propelled to the target position, respond to changes in external temperature and magnetic fields, and provide feedback on the perceived environmental temperature via radio frequency, which exhibit more capabilities and enhanced intelligence. Furthermore, the proposed manufacturing method for the robot is versatile and sensing modules can be extended from temperature to humidity, light, chemistry, and other areas, laying the solid foundation for future biomedical applications. However, in-depth research is needed when regarding the robot's actuation and retrieval after detection, and there are some problems required to be solved before the acceptance.

Response:

We are thankful to the reviewer for the positive assessment of our work and for the insightful suggestions that helped clarify the paper's contributions. We have carefully addressed each comment, refining the manuscript to emphasize our core innovations in integrating flexible electronics with soft microrobots to achieve actuation, sensing, and communication at a small scale. Through these revisions, we provide a clearer account of the robust integration method and remote communication mechanism, while also acknowledging current limitations and avenues for future exploration. We greatly appreciate the reviewer's detailed feedback, which has strengthened and sharpened our presentation of this research.

Comment 1: In the introduction, the authors mentioned that small-scale robots have made progress in coding and multimodality. The magnetic programming, anisotropic programming, and multi-function integration should be noted, just as exhibited in Sci. Adv.8, eabn8932(2022) and Nat Commun 10, 4087 (2019).

Response:

We appreciate the reviewer's emphasis on magnetic programming, anisotropic programming, and multi-function integration in small-scale robotics. In response, we have revised our Introduction to incorporate these important aspects, including references to the suggested studies (Sci. Adv. 8, eabn8932 (2022) and Nat. Commun. 10, 4087 (2019)), and expanded the discussion to highlight how such programming strategies and alignments provide multifunctional capabilities in microrobots.

page 3 (lines 50-51)

By incorporating anisotropic magnetic designs, researchers have introduced improved navigation and previously unobtainable functionalities into versatile microrobotic platforms^{8, 17, 19, 21, 30}.

Comment 2: In the part of the communication experiment, it was noted that an experimental platform exists between the transmitter and the receiver in addition to the robot. Whether the experimental platform affect the experimental results?

Response:

We agree with the reviewer's concern regarding the potential influence of the experimental platform on our RF communication experiments. In response, we conducted additional tests with different configurations of platform and media to quantify any variations in amplitude or trends of RF signals. Our findings indicate that while the holder can introduce minor amplitude shifts, these changes do not alter the overall signal trends and minor amplitude shifts can be subtracted via data processing. Therefore, the configurations of experimental platform do not affect to the experimental results. We have added these results to the manuscript.

Page 12 (lines 283-285)

The performance of RF communication-based shape detection can be significantly influenced by environmental factors, including the medium (e.g., ion-containing aqueous solutions) and the surroundings (e.g., experimental setups).

Page 13 (lines 308-319)

We also examined the effect of the experimental setup as a method to investigate surrounding influences. In the setup, the sample holder (i.e., a plate to place microrobots in the experimental setup) can affect the measured RF signals, as it is positioned between the antennas and can modulate signals. Note that we excluded effects of antennas, VNA, and cables, as they are pre-calibrated, making their contributions to signal variability negligible. First, we compared RF signals in two configurations of the experimental setup, that is, with and without the sample holder. As a result, overall trends of RF signals were similar across both experiments, while slight amplitude shift was observed (Supplementary Fig. 14a). Additionally, RF signals from different sample holder designs also show negligible effects (Supplementary Fig. 14b), demonstrating consistent shape detection performance across different experimental setup configurations. These findings indicate that while the sample holder may introduce minor amplitude deviations, it does not cause additional signal artifacts (e.g., extra peaks or dips). Any small shifts caused by the setup can be readily corrected during data processing.

Supplementary Fig. 14. Effect of the sample holder on radio communication. a. RF communication signals recorded with and without the holder. **b.** RF communication signals recorded using a well-type holder and a plate-type holder. While the overall trends remain similar across different conditions, slight amplitude shifts are observed. These signals are superimposed on background noise and post-processed to isolate the environmental effect.

Comment 3: New experiment environment should be designed to make the control of robot form a closed loop for better reflecting the advantages of magnetic field actuation and the importance of communication when the robot cannot be directly observed.

Response:

We are grateful to the reviewer for suggesting a closed-loop experiment for situations where direct observation is not feasible, as it could better highlight the advantages of both magnetic actuation and RF communication. In response, we performed additional experiments to demonstrate microrobot localization on a four-by-three array platform. This demonstration confirms that RF communications can detect the microrobots' position even under a shielding cover, reinforcing the potential of closed-loop control to enhance microrobot functionality through remote sensing and tracking in scenarios with limited visibility. We have revised the manuscript to include the result of this additional experiment.

Page 14 (lines 321-332)

Microrobot localization

Wireless communication capability enables microrobot localization and local information perception. Figure 4a illustrated that planar microrobots can be localized through areal scanning using transmitter and receiver antennas. This antenna setup scanned a sample holder divided into a four-by-three array and detected a notable dip (at ~12.2 GHz) in the measured RF signal at coordinate position (2, 3), where the planar microrobots were positioned (Fig. 4b; see also Supplementary Fig. 15a).

Additionally, when the microrobots were placed under a cover (Supplementary Fig. 15b,c), the planar and helical microrobots transmitted distinguishable RF signals, enabling shape detection (Fig. 4c). This demonstrates that planar microrobots can be detected via RF communication without the need for

a vision system. Finally, these demonstrations imply that developed microrobots can be localized even when covered, allowing their use for a local shape sensing.

Page 17 (lines 393-402)

Continuous tracking of the microrobot's shape demonstrates the feasibility of this approach for real-time, remote monitoring of the microrobot's morphological state. This enables remote recognition of the microrobot's configuration (e.g., helical vs. planar or folded vs. unfolded) during operation, addressing one of the key challenges in the practical use of microrobots. For example, configuration recognition can be advantageous in targeted delivery applications, where active shape reconfiguration (e.g., via magnetic hyperthermia induced by alternating magnetic fields) is required to adapt to complex environments. In such cases, remote communication can inform successful shape transformation, allowing the microrobot to proceed to its next task. Additionally, our microrobot localization experiments demonstrate that the system can reliably detect both position and shape, suggesting its potential for local temperature sensing applications.

Figure 4. RF communication-based microrobot localization. *a.* Schematic of RF communication-based localization, consisting of a scanning system and a sample holder divided into four-by-three array. The localization system consists of transmitter and receiver coils positioned opposite to each other, with scanning performed along a predefined path. *b.* RF signals across the array exhibit a distinctive signal at position (3,2) corresponding to the location of the planar microrobots. *c.* RF signals from seven units of 2×15 microrobots placed behind a barrier. The clearly distinct signals between helical and planar shapes demonstrate that the microrobots can be detected through the barrier.

Supplementary Fig. 15 Localization setup and shielded experimental setup. *a.* The experimental platform, divided into a 4×3 array. Two transmission coils were connected to the VNA, and the microrobots were positioned at (3,2). *b.* A shielded paper cover was placed between the microrobots and the transmitter coil, with an enlarged view shown in (c).

Comment 4: The author proposed that different types of electronics may be involved in the manufacturing process to highlight the scalability of the manufacturing method. Will these different electronics have variable signal effects during communication? If there are different responses, does it mean that the communication process is able to recognize multiple robots with different designs? Corresponding experiment could be designed to demonstrate this point.

Response:

We are thankful to the reviewer for the valuable advice regarding the varying communication performance across different types of electronics. In response, we further analyzed the RF signals from the dipole antennas of different sizes. We figured out that microrobots (i.e., dipole antennas) of different sizes (e.g., $2 \text{ mm} \times 15 \text{ mm}$ and $2 \text{ mm} \times 10 \text{ mm}$) exhibited a resonant frequency shift of $\sim 300 \text{ MHz}$, indicating that design parameters can vary the signal response. Additionally, we performed a finite element method simulation to investigate the resonant frequency of alternative designs, such as spiral antennas. This simulation also shows the resonant frequency tuning capability through design change. Accordingly, these suggest that antenna design can be leveraged to recognize multiple microrobots (or multiple sets of the same design microrobots). Meanwhile, we would like to clarify that we optimized the dipole antenna design to a size of $2 \text{ mm} \times 15 \text{ mm}$ to ensure better applicability within our demonstration scenario. We have revised the manuscript accordingly.

Page 16 (lines 375-382)

Additionally, we explored various pathways for improving the functionality of the developed microrobots. For example, different antenna designs produced measurably shifted resonant frequencies. Signals from two types of microrobots ($2\text{ mm} \times 15\text{ mm}$ and $2\text{ mm} \times 10\text{ mm}$ with a 10 cm gap between the transmitter and receiver antennas) displayed a difference in resonant frequency shifts of $\sim 300\text{ MHz}$ (Supplementary Fig. 12a(iv) and 12b(iv)). Furthermore, S_{21} varied with different antenna designs. As shown in Supplementary Fig. 17, the square spiral antenna exhibited a resonant frequency of $\sim 20.6\text{ GHz}$. Taken together, it becomes clear that careful antenna design can be used as a tool to distinguish microrobots, which when used in denser distributions, could enhance the signal sensitivity, as shown in Supplementary Fig. 18.

Supplementary Fig. 17. Resonant frequency of the new spiral antenna in air. The resonant frequency of the spiral antenna is $\sim 20.6\text{ GHz}$.

Supplementary Fig. 18. Effect of microrobot distribution density on RF communication using COMSOL simulation. A 33% increase in microrobot distribution density resulted in a threefold enhancement of the S_{21} amplitude at the resonant frequency. (Unit area: 1 cm^2)

Comment 5: In the final experiment, the robot is flattened due to the increase of temperature and the signal is captured by an external device, whether the flattened robot continue to respond to external magnetic field for subsequent recovery?

Response:

We are thankful to the reviewer for raising the concern regarding the subsequent recovery of planar microrobots. While rotating magnetic fields are typically preferred to generate rotational or

corkscrew motion in microrobots, gradient magnetic fields can also be employed to actuate the microrobots through dragging motion. In this study, the corkscrew locomotion using a helical shape was selected due to its advantage in achieving precise control. Nevertheless, magnetic field gradients can be used for magnetic guidance of microrobots. To address this point, we performed additional experiments demonstrating magnetic guidance of planar microrobots using gradient magnetic fields. These confirmed magnetic navigation capability for two different sizes of planar microrobots. We have included these results and revised the manuscript accordingly.

Page 8 (line 177-180)

Note that microrobots can also be manipulated in dragging motion using gradient fields, which can particularly be useful for controlling planar shape without relying on corkscrew motion capability (see Supplementary Fig. 7 and Movie S4). The average speed of $2\text{ mm} \times 15\text{ mm}$ microrobot was 1.2 mm s^{-1} under a magnetic field of 10 mT and gradient of 120 mT mm^{-1} .

Supplementary Fig. 7. Navigation of planar microrobots using a magnetic gradient field. A $2\text{ mm} \times 15\text{ mm}$ planar microrobot (a) and a $2\text{ mm} \times 5\text{ mm}$ planar microrobot (b) were guided by the magnetic dragging force. The magnetic field and gradient were set to 10 mT and 120 mT mm^{-1} , respectively. The average speeds of $2\text{ mm} \times 15\text{ mm}$ and $2\text{ mm} \times 5\text{ mm}$ microrobot were 1.2 mm s^{-1} and 0.7 mm s^{-1} , respectively (Scale bar: 10 mm).

Reviewer 2:

General comments: This manuscript presents a magnetically guided soft microrobot with integrated communication and environmental sensing capabilities. A proof of concept was demonstrated using a remotely controlled temperature-sensing microrobot. There have been numerous studies employing NIPAM-polymer for transformation and temperature sensing, rendering this work lacking in novelty. Consequently, I am unable to recommend this paper for publication in Nature Communications. Furthermore, there are several issues within the manuscript that require attention and revision by the authors:

Response:

We appreciate the reviewer's thorough evaluation of our work. However, we partially disagree with the concern regarding the novelty of our research. While previous studies have employed NIPAM hydrogels for temperature sensing, they typically rely on tethered systems, such as using electric wires (e.g., doi.org/10.1016/j.orgel.2020.105818) or optical fibers (e.g., doi.org/10.1021/acsami.7b00049). These systems inherently constrain the mobility and practical deployment of microrobots due to their dependence on external connections. Additionally, although NIPAM hydrogels have been utilized in microrobots for temperature sensing via photonic gel colorimetric responses (e.g., doi.org/10.1002/aisy.202100248), these methods face significant challenges in practical applications, particularly in achieving data readout in non-transparent environments. In contrast, our study presents a wireless, radio-communication-based temperature sensing microrobot that integrates flexible electronics with shape-morphable soft microrobots. To our knowledge, this represents the first demonstration of a fully untethered soft microrobot that combines instant environmental sensing with wireless communication, all while maintaining high mobility and adaptability.

In response to the reviewer's feedback, we have revised the manuscript to more clearly highlight our novel incorporation of flexible electronics for real-time communication and environmental sensing, thereby distinguishing our work from earlier efforts aimed at enhancing microrobot intelligence. Furthermore, we have conducted additional experiments and refined our discussion to better articulate the practical applications and significance of this technology. We are grateful for the reviewer's comments, which have greatly helped us strengthen both the clarity and rigor of our manuscript.

Comment 1: In the section reviewing related work, the manuscript mentions previous studies but fails to provide an in-depth comparative analysis of these works. A clearer discussion of how this study builds on or improves upon existing research is needed to emphasize its innovation and advantages.

Response:

We are thankful to the reviewer for highlighting the need for a clearer comparison with existing work. In response, we have revised the introduction to underscore our core innovation: the seamless

integration of flexible electronics into soft microrobots, which preserves inherent flexibility while providing a versatile platform for added functionalities on soft microrobots. By leveraging the direct conversion of smart material properties into readily interpretable electrical signals for remote communication, our approach not only addresses the limitations of more rigid or tethered designs but also establishes a robust framework for future enhancements, thereby advancing beyond the current state of the art in wireless, multifunctional microrobot research.

Page 3 (lines 50-54)

By incorporating anisotropic magnetic designs, researchers have introduced improved navigation and previously unobtainable functionalities into versatile microrobotic platforms^{8, 17, 19, 21, 30}. These individual microrobots have further advanced to multiscale¹², multimodal^{14, 15}, multi-agent³¹, hierarchical^{32, 33}, self-organizing^{22, 34}, and swarm behaviors^{13, 14, 35, 36}. And in doing so, they have expanded their applications in environmental remediation³⁷, micromanipulation³⁸, medicine^{39, 40}, and sensing⁴¹ contexts.

Page 4 (lines 63-100)

The integration of these technologies into microrobots became possible in the last few decades due to advances in microfabrication techniques, including those commonly used in microelectromechanical systems (MEMS)^{42, 43, 44} and complementary metal-oxide-semiconductor (CMOS)^{28, 29, 45}, alongside emerging methods such as 3D direct laser writing^{7, 24, 27, 46}. For example, Bandari et al. used microfabrication to create inductively powered microrobots with a strain engineering based form of chemical propulsion²⁹, and Miskin et al. integrated photovoltaics and surface electrochemical actuators for a light powered microrobot²⁸. While advanced fabrication techniques have opened up new possibilities for actuation methods, these implementations still struggle with a critical capability for real practical utility. Namely, these microrobots cannot communicate real-time information about their environment with external systems.

Roboticians have attempted to solve this problem in numerous fashions, whether by incorporating commercial antenna chips (including RFID) or through tethered approaches. These methods, based on readily available, low cost, and robust transmission technologies, can relay important information about their environment to external agents. For example, Li et al. demonstrated radiofrequency (RF)-capable, magnetically navigated microrobots that wirelessly transmitted temperature and pH data⁴⁷. However, both the chips and required auxiliary circuits force fundamental tradeoffs between signal quality and microrobot size while also requiring a rigid structure. Han and colleagues addressed some of these issues by using microfabrication and compressive buckling techniques to incorporate a flexible electronic sensor array within a balloon catheter that measured pressure and temperature⁴⁸. The soft nature of the flexible system better matches the mechanical properties seen in biology, but the tethered

setup still restricts the working range. As such, the question remains whether an untethered can successfully integrate the key aspects of remote navigation, collective behavior, and wireless sensing into a single system while maintaining the key advantage of flexible soft microrobots.

This research seeks to answer that question with a new microrobotic approach that, as its key innovation, integrates flexible electronics and a shape reconfigurable soft microrobot into a single device. This leverages smart materials to create large, physical changes in the microrobot in response to local stimuli and a flexible electronic antenna design that can take advantage of the entire microrobotic surface to propagate its signal. By coupling the two together, the physical changes to the robotic structure produce equally dramatic changes in the antenna signal character, enabling instant remote communication within a fully flexible microrobotic system. To achieve this, we developed an integration protocol to combine an anisotropic SU-8 passive layer with an iron oxide nanoparticle (IONP) embedded thermally-responsive hydrogel active layer, leading to a temperature dependent helical-to-planar transformation function. We then laminated a flexible dipole antenna for radio communication, completing the microrobot. In this work, we explain the new fabrication route and demonstrate microrobot magnetic navigation, RF communication-based shape detection, localization, and remote temperature sensing. We also show that the collective behavior of multiple microrobots enhances both RF signal sensitivity and reliability.

Comment 2: The temperature within the human body remains relatively stable, making it unlikely that a soft robot with temperature sensing capabilities would be necessary for human applications. Additionally, the size of the robot appears to be too large for such purposes. This prompts the question of which specific environment would require a soft robot with temperature sensing capabilities, further questioning its intended purpose. Additionally, it is unclear why magnetic control is necessary for this device.

Response:

We are thankful for the reviewer's concern. First, we would like to clarify the purpose behind the development of this microrobot. In many microrobotic applications, real-time recognition of the microrobot's state is challenging due to its small size and the limited resolution of conventional localization/imaging tools. For example, one important aspect is detection of the shape configuration (e.g., helical vs. planar) in targeted delivery applications. To navigate complex environments, the microrobot may need to actively change its shape (e.g., via programming its physical/chemical behavior or on-command by applying an external energy source, i.e., alternating magnetic fields to induce magnetic hyperthermia of IONP-hydrogel composites). Accordingly, it is important to ensure that the shape transformation has been accomplished before proceeding to the next task. To address this

challenge, our work proposes the integration of a flexible antenna into a soft microrobot, capable of real-time shape detection through remote communication.

Additionally, the shape detection functionality can be extended to temperature sensing by leveraging passive shape changes induced by environmental temperature variations. For example, this approach could be applied in human body environments to monitor localized temperature changes. While the overall body temperature remains relatively stable, localized temperature variations can occur. For example, temperature elevations of 0.5-2 °C near tumor sites and 1-3 °C in regions of inflammation have been reported. Additionally, the size of the microrobot is compatible with anatomical dimensions typically found in the gastrointestinal (GI) tract, supporting its potential feasibility for biomedical applications in such environments. As magnetic fields are widely used to actuate microrobots across barriers, we employed this method for microrobot control.

To better communicate our findings on the microrobot's shape recognition functionality, we have revised the manuscript.

Page 2 (lines 25-31, abstract)

As a proof of concept, we present a microrobot, which integrates a thermoresponsive magnetic hydrogel, an anisotropic support structure, and a flexible dipole antenna into a cohesive three-layered design. The microrobot can morph from helical shape at low-temperatures to planar shape at high-temperatures. This shape transformation can be remotely detected by external radio communication receivers, enabling shape-state recognition and environmental temperature sensing. Furthermore, we show that the collective behavior of multiple microrobots enhances the recognition performance by amplifying the signal.

Page 4 (lines 84-86)

As such, the question remains whether an untethered can successfully integrate the key aspects of remote navigation, collective behavior, and wireless sensing into a single system while maintaining the key advantage of flexible soft microrobots.

Page 15 (lines 356-361)

We demonstrated remote communication with soft, shape-reconfigurable microrobots, featuring a key innovation in the integration of a flexible dipole antenna. The microrobot's temperature-responsive shape-changing ability directly translates into antenna reconfiguration, providing a clear mechanism for detecting its morphological state remotely. By designing the antenna to match the microrobot's dimensions, we significantly enhance the sensitivity and reliability of radio communication signals. This approach overcomes challenges in poor signal sensitivity common in previous attempts with similar micro antennas.

Page 17 (lines 393-418)

Continuous tracking of the microrobot's shape demonstrates the feasibility of this approach for real-time, remote monitoring of the microrobot's morphological state. This enables remote recognition of the microrobot's configuration (e.g., helical vs. planar or folded vs. unfolded) during operation, addressing one of the key challenges in the practical use of microrobots. For example, configuration recognition can be advantageous in targeted delivery applications, where active shape reconfiguration (e.g., via magnetic hyperthermia induced by alternating magnetic fields) is required to adapt to complex environments. In such cases, remote communication can inform successful shape transformation, allowing the microrobot to proceed to its next task. Additionally, our microrobot localization experiments demonstrate that the system can reliably detect both position and shape, suggesting its potential for local temperature sensing applications.

The used hydrogels in this study swell at the 34-37 °C of lower critical solution temperature (LCST) with a narrow LCST range⁶², which can be further tuned by adding hydrophilic and hydrophobic monomers and/or modifying the polymer architecture. By manipulating the swelling behavior, the detectable temperature range can be varied⁶³. Based on this, the temperature sensing accuracy, which is originally limited by the intrinsic nature of hydrogel-based temperature sensing, may be further improved through fuzzy logic approach, leveraging the collective effect of microrobots designed with varying phase change temperature. Additionally, the type of hydrogels can be extended to various stimulus-responsive hydrogels (e.g., pH, light, electric, chemical) beyond the heat-detection demonstrated here. Thus, this microrobotic concept has the potential to be expanded into a wide range of devices for monitoring diverse stimuli at the microscale.

Taking the remotely controlled actuation and local environmental sensing together, one begins to see an interesting use case for the developed microrobotic platform in biomedical applications (e.g., binary temperature sensing in localized diagnostic applications. Regions of tumor or inflammation exhibit higher temperatures, typically elevated by 0.5–3 °C).

Page 18 (lines 430-431)

Combined, this wireless communication-capable microrobotic platform holds promise for future medical use.

Page 19 (lines 441-449)

In summary, our proposed microrobot design and manufacturing protocol, which integrates flexible electronics with soft microrobots, enables the development of fully flexible, wireless-sensing microrobots capable of magnetic navigation and collective behavior. The ultra-thin dipole antenna remains fully functional under the mechanical strain induced by shape transformation of the microrobots, transmitting different radio communication signals between planar and helical shapes.

The collective behavior further enhances the signal differentiation, improving remote shape detection and ensuring reliability in RF-based temperature sensing. We expect that this advancement could open new possibilities in embodied micro-intelligence that can interact to environment and gather data, thereby paving the way for the autonomous operation of small-scale robots.

Comment 3: The robot undergoes deformation in response to temperature stimuli. Is the degree of deformation consistent across trials? How much unexpected variation in deformation impacts the accuracy of the sensing signal?

Response:

We appreciate the reviewer's inquiry regarding the consistency of the microrobot's deformation and its impact on sensing accuracy. To address this, we additionally evaluated the dimensions of the planar and helical microrobots across the temperature variation, figuring out that the shape-morphing was achieved with minimal variation. This consistency supports our focus on a binary communication paradigm, where differentiation between the planar and helical configurations is the significant factor. We have revised the manuscript accordingly to include these findings and clarify the deformation consistency.

Page 8 (lines 185-194)

Additionally, the repeatability of shape reconfiguration was evaluated by analyzing the average body length and its standard deviation. We found that the average body length of planar microrobots varied from 14.9 mm to 14.4 mm over two rounds of shape morphing, while that of helical microrobots ranged from 11.4 mm to 10.9 mm. When the average length of the planar microrobots is divided by designed body length, it results in an error of ~0.6-4%, indicating that shape sensing is repeatable. Furthermore, the consistency of shape morphing among multiple microrobots was assessed using the standard deviation-to-average ratio (i.e., coefficient of variation) of body length across three microrobot units. The planar shape shows coefficient of variation values of ~0.038 and ~0.063 over two rounds of shape morphing, indicating a stable shape reconfiguration across the microrobots in response to environmental temperature changes.

Comment 4: Deformation is triggered when the temperature exceeds a certain threshold. With such conditions, how can the robot be deployed into a real system?

Response:

We are thankful to the reviewer for this insightful suggestion. To clarify the conditions, we performed additional experiments to analyze shape deformation as a function of temperature. The hydrogels used in this research have 34-37 °C of lower critical solution temperature (LCST), while our analysis identified the phase change temperature to be approximately 40 °C. Based on these results, we

may propose a potential use-case scenario in which the microrobots can be deployed in environments with temperatures not exceeding 43 °C, with the aim of detecting whether the local temperature surpasses 39 °C (for example, in localized temperature sensing within human body).

Additionally, in our other study, we observed that variation in bilayer structures (e.g., using [the same hydrogel]/[SU8] with different thickness combinations or using [different hydrogel]/[SU8] with the same thickness combinations) can increase the phase change temperature (e.g., up to 60 °C). Also, it is well known that the LCST can be tuned by the hydrogel types (doi.org/10.1021/acs.langmuir.5b02948, doi.org/10.1039/D3LP00114H). Thus, by tuning the phase change temperature, the microrobots can be deployed in environments requiring local binary temperature sensing, such as temperature-responsive micromanipulation, environmental monitoring, and diagnostic temperature-sensing in medical applications.

Within these tunable temperature ranges, microrobots can be deployed in microrobotic applications discussed in our Response to Comment 2. We have revised the manuscript accordingly.

Page 15 (lines 342-346)

Additionally, the shape deformation from helical to planar was evaluated as a function of temperature (Fig. 5d). Note that the phase change temperature was set to 40 °C, as the shape deformation from helical to fully planar was completed within a reasonable timeframe (less than 10 seconds) under this condition (Supplementary Fig. 16). These deformation results were then correlated with the RF signal amplitude at the resonant frequency.

Page 17 (lines 393-418)

Continuous tracking of the microrobot's shape demonstrates the feasibility of this approach for real-time, remote monitoring of the microrobot's morphological state. This enables remote recognition of the microrobot's configuration (e.g., helical vs. planar or folded vs. unfolded) during operation, addressing one of the key challenges in the practical use of microrobots. For example, configuration recognition can be advantageous in targeted delivery applications, where active shape reconfiguration (e.g., via magnetic hyperthermia induced by alternating magnetic fields) is required to adapt to complex environments. In such cases, remote communication can inform successful shape transformation, allowing the microrobot to proceed to its next task. Additionally, our microrobot localization experiments demonstrate that the system can reliably detect both position and shape, suggesting its potential for local temperature sensing applications.

The used hydrogels in this study swell at the 34-37 °C of lower critical solution temperature (LCST) with a narrow LCST range⁶², which can be further tuned by adding hydrophilic and hydrophobic monomers and/or modifying the polymer architecture. By manipulating the swelling behavior, the detectable temperature range can be varied⁶³. Based on this, the temperature sensing accuracy, which

is originally limited by the intrinsic nature of hydrogel-based temperature sensing, may be further improved through fuzzy logic approach, leveraging the collective effect of microrobots designed with varying phase change temperature. Additionally, the type of hydrogels can be extended to various stimulus-responsive hydrogels (e.g., pH, light, electric, chemical) beyond the heat-detection demonstrated here. Thus, this microrobotic concept has the potential to be expanded into a wide range of devices for monitoring diverse stimuli at the microscale.

Taking the remotely controlled actuation and local environmental sensing together, one begins to see an interesting use case for the developed microrobotic platform in biomedical applications (e.g., binary temperature sensing in localized diagnostic applications. Regions of tumor or inflammation exhibit higher temperatures, typically elevated by 0.5–3 °C).

Figure 5. Demonstration of magnetic navigation, thermoresponsive shape reconfiguration, and remote-temperature sensing with three microrobots. d. Shape deformation from helical (0%) to planar (100%) as a function of PBS temperature. The signal amplitude at resonant frequency is estimated from the shape.

Comment 5: In complex environments, such as natural or living settings, how can the sensitivity and accuracy of the sensor signals be ensured?

Response:

We are thankful to the reviewer for highlighting the importance of sensing performance in complex environments. First, we would like to clarify that the demonstration of our microrobots in natural or living settings is beyond scope of this work. However, we agree with that the sensitivity and accuracy of the sensing signals should be evaluated in more complex, realistic environments.

Accordingly, we conducted additional experiments and finite element method simulations to evaluate shape detection performance in different media (e.g., phosphate-buffered saline (PBS), which closely mimics the ionic composition of body fluids). Theoretical analysis, computational simulations,

and experimental measurements confirm that the resonant frequency of the planar antenna is observed with a shift from ~12.0 GHz in air to ~1.5 GHz in ion-rich media. These results indicate that our system can maintain reliable sensing and communication for distinguishing between planar and helical shapes in more naturalistic environments.

We have revised the manuscript accordingly to include these results. Additionally, we have clarified that our work is not solely focused on designing a temperature-sensing microrobot for biomedical applications and have removed any potentially confusing references in the manuscript.

Page 12 (lines 282-306)

Investigation of environmental effects

The performance of RF communication-based shape detection can be significantly influenced by environmental factors, including the medium (e.g., ion-containing aqueous solutions) and the surroundings (e.g., experimental setups). First, different media can shift the resonant frequency of RF signals. The basic mechanism of the dipole antenna's resonance is based on the matched wavelength of the electromagnetic wave. The wavelength at the resonant frequency is defined as:

$$\lambda = \frac{c}{f_{r,air}} \quad (5)$$

where c and $f_{r,air}$ are the speed of light and the resonant frequency in air, respectively. In a specific medium, the speed of light v_m is given by:

$$v_m = \frac{c}{n} \quad (6)$$

where n is the refractive index of the medium. Based on Maxwell's electromagnetic field theory, the refractive index is given by $n = \sqrt{\epsilon_r \mu_r}$. For non-magnetic materials, the relative magnetic permeability index μ_r is approximately 1, meaning that the refractive index depends solely on the relative dielectric constant ϵ_r . Thus, the refractive index simplifies to $n = \sqrt{\epsilon_r}$. Since the electromagnetic wavelength for the resonance of dipole antenna needs to remain the same across different media (e.g., air and ion-containing aqueous solutions), Equation (5) and (6) can be rewritten as:

$$\frac{c}{f_{r,air}} = \frac{v_m}{f_{r,medium}} = \frac{c/n}{f_{r,medium}} \quad (7)$$

where $f_{r,medium}$ is the resonant frequency of the dipole antenna in a specific medium. Thus, the resonant frequency $f_{r,medium}$ is defined by Equation (8).

$$f_{r,medium} = \frac{f_{r,air}}{\sqrt{\epsilon_r}} \quad (8)$$

Considering the developed 2 mm × 15 mm microrobots, they exhibited a resonant frequency of ~12 GHz in air. When immersed them in phosphate-buffered saline (PBS, a relative permittivity: ~70), the resonant frequency is expected to shift to ~1.45 GHz. Figure 3h shows that experimentally measured resonant frequency shifts to ~1.52 GHz in the PBS medium, confirming above analytical estimation is

reasonable. This analytic estimation and experimental results also aligned with a simulation result, which shows a resonant frequency shifts to ~ 1.47 GHz (Supplementary Fig. 13).

Page 16 (lines 384-391)

We also theoretically and experimentally confirmed that the use of microrobots in the ion-containing aqueous solution medium significantly lowered the resonant frequency and enhanced the RF communication performance, where one-order reduction in resonant frequency enhances the penetration capability of RF signals. Comparing the RF signals of the dipole antenna in PBS at 2 mm and 4 mm depths shows a clearer signal distinction between signals at a greater depth. (Supplementary Fig. 19). This is probably because the increased medium thickness absorbs more RF signals⁶¹, enhancing signal sensitivity. Additionally, since human body fluids are similarly ion-containing aqueous solutions, there is potential applicability for these microrobots in biomedical applications.

Figure 3. Design parameter study and shape detecting performance through RF communication. h. RF signal shifts according to shape reconfiguration from helical to planar of seven units of 2×15 microrobots immersed in PBS solution. The resonant frequency of the microrobots in PBS solution shifts to ~ 1.5 GHz, whereas ~ 1.2 GHz in air.

Supplementary Figure 13. Simulation of medium effects on RF communication. The COMSOL simulation with a PBS medium shows a resonant frequency shift to a lower region, consistent with experimental results.

Supplementary Fig. 19. Experimental comparison of microrobots' RF signal response at the resonant frequency in PBS. As the depth increases, communication performance improves, leading to a more significant absolute amplitude difference.

Comment 6: While using multiple robots enhances the sensing signal, the impact of inter-robot distance and relative orientation on signal quality requires further evaluation.

Response:

We are thankful to the reviewer for this insightful suggestion. We agree that the distance and relative orientation of the multiple microrobots can influence radio communication performance. Theoretically, a denser distribution of microrobots is expected to be more effective, as a higher number of microrobots leads to increased signal amplitude.

To support this, we performed finite element method simulations comparing the signal strength between two different microrobot densities. In result, the array with 33% higher density showed a threefold increase in signal difference, reaching to -30 dB (as shown in supplementary Fig. 17). These findings are consistent with our experimental results observed in collective microrobot behaviors, further validating the impact of microrobot density on communication performance.

Regarding microrobot orientation, we acknowledge that RF signal transmission and reception are dependent on the relative orientations of the transmitter, receiver, and antenna. However, in our system, the relative orientation of microrobots can be actively controlled via an external magnetic field, ensuring alignment across all units. Therefore, we believe similar results would be achieved in practical applications where uniform orientation is maintained. We have revised the manuscript accordingly.

Page 16 (lines 380-382)

Taken together, it becomes clear that careful antenna design can be used as a tool to distinguish microrobots, which when used in denser distributions, could enhance the signal sensitivity, as shown in Supplementary Fig. 18.

Supplementary Fig. 18. Effect of microrobot distribution density on RF communication using COMSOL simulation. A 33% increase in microrobot distribution density resulted in a threefold enhancement of the S_{21} amplitude at the resonant frequency. (Unit area: 1 cm^2)

Comment 7: The simplified experimental conditions used in this study may not fully capture the challenges and limitations that microrobots could encounter in real-world applications. A broader range of scenarios should be tested to assess these factors more comprehensively.

Response:

We are thankful to the reviewer for emphasizing the need to evaluate a broader spectrum of operational scenarios. In response, we conducted additional experiments to demonstrate the microrobots' remote localization and signal detection capabilities, including tests under a shielding cover and across a three-by-four array platform. These experiments serve as proof-of-concept for the functionality of our integrated flexible electronics and soft microrobot system in more complex and less accessible environments, thereby offering an initial foundation for further exploration of real-world applications. We have revised the manuscript to include these results.

Page 14 (lines 321-332)

Microrobot localization

Wireless communication capability enables microrobot localization and local information perception. Figure 4a illustrated that planar microrobots can be localized through areal scanning using transmitter and receiver antennas. This antenna setup scanned a sample holder divided into a four-by-three array and detected a notable dip (at $\sim 12.2 \text{ GHz}$) in the measured RF signal at coordinate position (2, 3), where the planar microrobots were positioned (Fig. 4b; see also Supplementary Fig. 15a).

Additionally, when the microrobots were placed under a cover (Supplementary Fig. 15b,c), the planar and helical microrobots transmitted distinguishable RF signals, enabling shape detection (Fig. 4c). This demonstrates that planar microrobots can be detected via RF communication without the need for a vision system. Finally, these demonstrations imply that developed microrobots can be localized even when covered, allowing their use for a local shape sensing.

Figure 4. RF communication-based microrobot localization. *a.* Schematic of RF communication-based localization, consisting of a scanning system and a sample holder divided into four-by-three array. The localization system consists of transmitter and receiver coils positioned opposite to each other, with scanning performed along a predefined path. *b.* RF signals across the array exhibit a distinctive signal at position (3,2) corresponding to the location of the planar microrobots. *c.* RF signals from seven units of 2×15 microrobots placed behind a barrier. The clearly distinct signals between helical and planar shapes demonstrate that the microrobots can be detected through the barrier.

Supplementary Fig. 15 Localization setup and shielded experimental setup. *a.* The experimental platform, divided into a 4×3 array. Two transmission coils were connected to the VNA, and the microrobots were positioned at (3,2). *b.* A shielded paper cover was placed between the microrobots and the transmitter coil, with an enlarged view shown in (c).

Comment 8: The manuscript is insufficient to verify the practical application of the robot. Although biomedical applications are mentioned, the challenges of operating in complex biological environments, as well as biocompatibility issues, are not sufficiently addressed. There is also a lack of discussion or data on potential health risks, such as immune responses or toxicity, that microrobots might pose in living organisms.

Response:

We appreciate the reviewer's concerns regarding the practical application and biocompatibility of our microrobots. While we do provide preliminary biocompatibility test results (Supplementary Fig. 19), our primary goal is to showcase the seamless integration of flexible electronics with soft microrobots rather than to develop a dedicated biomedical device at this stage. To address potential challenges in complex biological settings, we conducted additional experiments in phosphate-buffered saline (PBS) with a similar ionic environment of body fluids. These results demonstrate our system's broader applicability and lay the groundwork for future investigations, including more extensive safety and biocompatibility assessments in realistic physiological conditions. We have revised the manuscript accordingly.

Page 12 (lines 281-306, Duplicated from Response 5 for the reviewer's reference.)

Investigation of environmental effects

The performance of RF communication-based shape detection can be significantly influenced by environmental factors, including the medium (e.g., ion-containing aqueous solutions) and the surroundings (e.g., experimental setups). First, different media can shift the resonant frequency of RF signals. The basic mechanism of the dipole antenna's resonance is based on the matched wavelength of the electromagnetic wave. The wavelength at the resonant frequency is defined as:

$$\lambda = \frac{c}{f_{r,air}} \quad (5)$$

where c and $f_{r,air}$ are the speed of light and the resonant frequency in air, respectively. In a specific medium, the speed of light v_m is given by:

$$v_m = \frac{c}{n} \quad (6)$$

where n is the refractive index of the medium. Based on Maxwell's electromagnetic field theory, the refractive index is given by $n = \sqrt{\epsilon_r \mu_r}$. For non-magnetic materials, the relative magnetic permeability index μ_r is approximately 1, meaning that the refractive index depends solely on the relative dielectric constant ϵ_r . Thus, the refractive index simplifies to $n = \sqrt{\epsilon_r}$. Since the electromagnetic wavelength for the resonance of dipole antenna needs to remain the same across different media (e.g., air and ion-containing aqueous solutions), Equation (5) and (6) can be rewritten as:

$$\frac{c}{f_{r,air}} = \frac{v_m}{f_{r,medium}} = \frac{c/n}{f_{r,medium}} \quad (7)$$

where $f_{r,medium}$ is the resonant frequency of the dipole antenna in a specific medium. Thus, the resonant frequency $f_{r,medium}$ is defined by Equation (8).

$$f_{r,medium} = \frac{f_{r,air}}{\sqrt{\epsilon_r}} \quad (8)$$

Considering the developed $2 \text{ mm} \times 15 \text{ mm}$ microrobots, they exhibited a resonant frequency of ~ 1.2 GHz in air. When immersed them in phosphate-buffered saline (PBS, a relative permittivity: ~ 70), the resonant frequency is expected to shift to ~ 1.45 GHz. Figure 3h shows that experimentally measured resonant frequency shifts to ~ 1.52 GHz in the PBS medium, confirming above analytical estimation is reasonable. This analytic estimation and experimental results also aligned with a simulation result, which shows a resonant frequency shifts to ~ 1.47 GHz (Supplementary Fig. 13).

Page 16 (lines 384-391, Duplicated from Response 5 for the reviewer's reference.)

We also theoretically and experimentally confirmed that the use of microrobots in the ion-containing aqueous solution medium significantly lowered the resonant frequency and enhanced the RF communication performance, where one-order reduction in resonant frequency enhances the penetration capability of RF signals. Comparing the RF signals of the dipole antenna in PBS at 2 mm and 4 mm depths shows a clearer signal distinction between signals at a greater depth. (Supplementary Fig. 19). This is probably because the increased medium thickness absorbs more RF signals⁶¹, enhancing signal sensitivity. Additionally, since human body fluids are similarly ion-containing aqueous solutions, there is potential applicability for these microrobots in biomedical applications.

Page 18 (lines 415-431)

Taking the remotely controlled actuation and local environmental sensing together, one begins to see an interesting use case for the developed microrobotic platform in biomedical applications (e.g., binary temperature sensing in localized diagnostic applications. Regions of tumor or inflammation exhibit higher temperatures, typically elevated by $0.5\text{--}3$ °C). As an initial step, we examined the feasibility of using this microrobot in bio-settings by testing the material biocompatibility with MTT assays (Supplementary Fig. 20). When exposed to human umbilical vein endothelial cell (HUVEC) cultures for 24 and 72 hours, the microrobots demonstrated over 80% of cell viability, indicating acceptable levels of biocompatibility. Meanwhile, studying radio communication in bio-settings reveals both promising aspects and ongoing challenges. For example, the penetration depth of RF electromagnetic waves (1-5 GHz) through tissues falls within a reasonable range (16-60 mm). However, signal reflection at the air-tissue interface remains a limitation, indicating the need for further optimization of external transmitter and receiver coil designs. In addition, while increasing transmitting coil power can help mitigate signal attenuation over these depths, human exposure to RF electromagnetic field should be considered. Although research is ongoing, power levels can be increased within safe limits defined by the tissue heating capacity, measured by the specific absorption rate (SAR), which is

generally accepted to be less than 2 W kg^{-1} for localized exposure.⁶⁴ Combined, this wireless communication-capable microrobotic platform holds promise for future medical use.

Figure 3. Design parameter study and shape detecting performance through RF communication. h. RF signal shifts according to shape reconfiguration from helical to planar of seven units of 2×15 microrobots immersed in PBS solution. The resonant frequency of the microrobots in PBS solution shifts to ~ 1.5 GHz, whereas ~ 12 GHz in air.

Supplementary Figure 13. Simulation of medium effects on RF communication. The COMSOL simulation with a PBS medium shows a resonant frequency shift to a lower region, consistent with experimental results.

Supplementary Fig. 19. Experimental comparison of microrobots' RF signal response at the resonant frequency in PBS. As the depth increases, communication performance improves, leading to a more significant absolute amplitude difference.

Supplementary Fig. 20. Biocompatibility of microrobots. Using MTT assays with human umbilical vein endothelial cells (HUVEC), the microrobots were subjected to cell culturing conditions for 24 and 72 hours, demonstrating sufficient biocompatibility.

Comment 9: The manuscript does not propose specific strategies or plans for mass production of the microrobots or for their integration into existing systems or processes. Guidance on how to scale production and facilitate practical deployment would be beneficial.

Response:

We are thankful to the reviewer for highlighting the importance of scalable fabrication. In response, we have provided a detailed fabrication protocol flowchart (Supplementary Fig. 21) to illustrate how our microrobots can be produced via established semiconductor manufacturing processes. By leveraging wafer-scale techniques, we demonstrate high-yield batch production, with each run achieving an average of 80% yield on a four-inch wafer. This approach underscores the intrinsic compatibility of our design with current industry-standard fabrication methods, facilitating reliable integration of flexible electronics into soft microrobots at scale.

The detailed fabrication protocol flow chart is depicted in the Supplementary Fig. 22 and written in the Supplementary Methods section.

Supplementary Fig. 22. Fabrication and integration workflow chart.

Reviewer 4:

General comments: This article reports on the fabrication and testing of a new class of mm scale lithographically fabricated robot which combines the functionalities of locomotion, temperature sensing, and data transmission via RF signals, in a single device platform. The key advance is the combination of a temperature sensitive bending magnetic hydrogel/SU8 bilayer with a gold antennae layer, to realize a trilayer device can sense and report information about temperature via deformation of the antennae by the hydrogel. The authors outline the concept of operation for the device, the fabrication protocol by which the devices are built, the effect that bending has on RF signal amplitude, how the effect increases with number of devices, and demonstrate that devices can locomote using external magnetic fields to regions of varying temperature. Overall, the biggest strength of this paper is taking up the task of integrating multiple functionalities into a single robot. The following changes would likely improve the quality of the manuscript:

Response:

We appreciate the reviewer's valuable assessment of our work and are grateful for the constructive suggestions provided. In response, we have conducted additional experiments and revised the manuscript to address each comment in detail. These results not only reaffirm our core findings but also clarify and highlight the novel integration of flexible electronics into the soft microrobots to achieve communication. We believe these revisions significantly strengthen the manuscript and enhance its overall readability.

Comment 1: Remote sensing generally denotes that one can infer the specific value of temperature from the action of the device. It would be a lot easier to judge this work if data were included that quantifies how the detected radio signal depends on environmental temperature. Specifically, we feel that somewhere in the main text there needs to be a graph that shows temperature vs the authors preferred measurement from the antennae (i.e. resonance amplitude, resonance frequency shift). All we could find was the statement that the hydrogel undergoes a phase transition around 35C, and the data in 4F where robots are stated to unfold at 40C. Depending on the width of this transition, these devices could report across biomedically relevant temperatures, or be completely unsuited as a body temperature of 40C constitutes a medical emergency. Some data about how temperature can be extracted and quantified based on the device behavior would dramatically improve this work. Better still would be a thoughtful discussion about device accuracy and precision, given this is a completely analog reporting system.

Response:

We are thankful to the reviewer for highlighting the need to clarify how the radio signal depends on environmental temperature. In response, we performed additional experiments to analyze shape deformation as a function of temperature. The hydrogels used in this research have 34-37 °C of lower

critical solution temperature (LCST), while our analysis identified the phase change temperature to be approximately 40 °C. The shape deformation results were then correlated with the RF signal amplitude at the resonant frequency.

We also would like to explain the purpose behind the development of this microrobot. In many microrobotic applications, real-time recognition of the microrobot's state is challenging due to its small size and the limited resolution of conventional localization/imaging tools. For example, one important aspect is detection of the shape configuration (e.g., helical vs. planar) in targeted delivery applications. To navigate complex environments, the microrobot may need to actively change its shape (e.g., via programming its physical/chemical behavior or on-command by applying an external energy source, i.e., alternating magnetic fields to induce magnetic hyperthermia of IONP-hydrogel composites). Accordingly, it is important to ensure that the shape transformation has been accomplished before proceeding to the next task. To address this challenge, our work proposes the integration of a flexible antenna into a soft microrobot, capable of real-time shape detection through remote communication.

Additionally, the shape detection functionality can be extended to temperature sensing by leveraging passive shape changes induced by environmental temperature variations. It needs to note that the temperature sensing accuracy of our system is lower than that of conventional digital sensors, due to the intrinsic nature of the hydrogel-based sensing mechanism. Therefore, the use-case scenario needs to be defined accordingly. For example, we may propose that these microrobots can be deployed in environments where temperatures do not exceed 43 °C, with the specific aim of detecting whether the local temperature surpasses 39 °C. While the overall body temperature remains relatively stable, localized temperature variations can occur. For example, temperature elevations of 0.5-2 °C near tumor sites and 1-3 °C in regions of inflammation have been reported. Thus, the microrobots may be well suited for binary temperature sensing in localized diagnostic applications.

Alternatively, fuzzy logic approaches could be applied using microrobots designed with varying phase change temperature. In our other study, we observed that variation in bilayer structures (e.g., using [the same hydrogel]/[SU8] with different thickness combinations or using [different hydrogel]/[SU8] with the same thickness combinations) can increase the phase change temperature (e.g., up to 60 °C). Also, it is well known that the LCST can be tuned by the hydrogel types (doi.org/10.1021/acs.langmuir.5b02948, doi.org/10.1039/D3LP00114H).

We have included these findings and the related discussion into the revised manuscript.

Page 2 (lines 25-31, abstract)

As a proof of concept, we present a microrobot, which integrates a thermoresponsive magnetic hydrogel, an anisotropic support structure, and a flexible dipole antenna into a cohesive three-layered design. The microrobot can morph from helical shape at low-temperatures to planar shape at high-temperatures. This shape transformation can be remotely detected by external radio communication receivers, enabling shape-state recognition and environmental temperature sensing. Furthermore, we

show that the collective behavior of multiple microrobots enhances the recognition performance by amplifying the signal.

Page 10 (lines 230-233)

This was demonstrated by continuously tracking the RF signals as a single microrobot's shape morphed from planar to helical (shown in Fig. 3d). During the shape transition, the amplitude at the resonance frequency gradually shifted by an order of 10^1 , highlighting the microrobot's capability for real-time shape sensing.

Page 15 (lines 334-353)

Remote temperature-sensing

To demonstrate the full process for thermal measurement via radio communication-based shape detection, we constructed a functional example of the process (Fig. 5 and Supplementary Movie S6). We positioned three helical microrobots (2 mm × 15 mm) in a water channel with varying temperatures, simulating the conditions of the targeted temperature-sensing application (Fig. 5a). The microrobots were actuated under rotating magnetic fields (40 mT at 4 Hz) using a magnetic navigation system (experimental configuration in Supplementary Fig. 6), propelled along the water channel from the low-temperature zone to the high-temperature zone via corkscrew motion (Fig. 5b). Upon reaching the high-temperature zone (~40 °C), the microrobots transformed from a helical to a planar shape (Fig. 5c). Additionally, the shape deformation from helical to planar was evaluated as a function of temperature (Fig. 5d). Note that the phase change temperature was set to 40 °C, as the shape deformation from helical to fully planar was completed within a reasonable timeframe (less than 10 seconds) under this condition (Supplementary Fig. 16). These deformation results were then correlated with the RF signal amplitude at the resonant frequency. Based on this analysis, we remotely detected the shape transformations (Fig. 5e,f) through RF signals using the VNA. Comparing the RF signal from these microrobots (Fig. 5g) with our existing data set, we could estimate whether the environmental temperature exceeded the phase change temperature. The pink line in Fig. 5g corresponds to the fully planar shape, indicating a temperature in the range of 39-41 °C. Note that experiments were performed sequentially for i) navigation and transformation and then ii) transformation and temperature sensing, as the Joule heating coils in the navigation setup can distort the radio communication signals.

Page 15 (lines 357-359)

The microrobot's temperature-responsive shape-changing ability directly translates into antenna reconfiguration, providing a clear mechanism for detecting its morphological state remotely.

Page 17 (lines 393-413)

Continuous tracking of the microrobot's shape demonstrates the feasibility of this approach for real-time, remote monitoring of the microrobot's morphological state. This enables remote recognition of the microrobot's configuration (e.g., helical vs. planar or folded vs. unfolded) during operation, addressing one of the key challenges in the practical use of microrobots. For example, configuration recognition can be advantageous in targeted delivery applications, where active shape reconfiguration (e.g., via magnetic hyperthermia induced by alternating magnetic fields) is required to adapt to complex environments. In such cases, remote communication can inform successful shape transformation, allowing the microrobot to proceed to its next task. Additionally, our microrobot localization experiments demonstrate that the system can reliably detect both position and shape, suggesting its potential for local temperature sensing applications.

The used hydrogels in this study swell at the 34-37 °C of lower critical solution temperature (LCST) with a narrow LCST range⁶², which can be further tuned by adding hydrophilic and hydrophobic monomers and/or modifying the polymer architecture. By manipulating the swelling behavior, the detectable temperature range can be varied⁶³. Based on this, the temperature sensing accuracy, which is originally limited by the intrinsic nature of hydrogel-based temperature sensing, may be further improved through fuzzy logic approach, leveraging the collective effect of microrobots designed with varying phase change temperature. Additionally, the type of hydrogels can be extended to various stimulus-responsive hydrogels (e.g., pH, light, electric, chemical) beyond the heat-detection demonstrated here. Thus, this microrobotic concept has the potential to be expanded into a wide range of devices for monitoring diverse stimuli at the microscale.

Figure 3. Design parameter study and shape detecting performance through RF communication. d. Real-time RF signal monitoring during continuous shape deformation from helical to planar, showing signal amplitude increase throughout the transition.

Figure 5. Demonstration of magnetic navigation, thermoresponsive shape reconfiguration, and remote-temperature sensing with three microrobots. d. Shape deformation from helical (0%) to planar (100%) as a function of PBS temperature. The signal amplitude at resonant frequency is estimated from the shape.

Supplementary Fig. 16. Time for shape deformation from helical to planar as a function of temperature. Note that the exact deformation time may vary depending on factors such as magnetic nanoparticle density, crosslinking density, and film thickness. However, the overall trend remains consistent.

Comment 2: 1-10 mm is a large length scale for microelectronics, allowing for a range of off-the-shelf sensing and communication modalities. For instance, the chips used to control commercial RFID antennas are typically just mm in size (i.e., ignoring the antenna, the control circuit that packages data can be well under the size of this robot). Likewise, there are numerous publications demonstrating CMOS temperature sensors and chips well under 1 mm in size. This begs the question: why this schema? Why not use an integrated circuit to measure and modulate RF signal directly? Or, for that matter, why not use acoustic communication, such as in neural dust? These alternatives would have significant benefits in signal reliability and accuracy; the authors should explain the merits of their approach, as compared to equally large devices.

Response:

We are thankful to the reviewer for the suggestion to include a discussion on alternative sensing and communication approaches, such as commercial RFID chips, integrated circuits, or acoustic methods. While we agree on the potential of microelectronics to expand the functionalities of microrobots, we believe their use is limited within the scope of our approaches for several reasons.

Flexibility (i.e., softness) is a desirable feature in microrobots, as it enables compatibility with environments of varying stiffness and allows the microrobots to conform to complex geometries. As demonstrated in our work, flexibility also enables shape-morphing capabilities that would be difficult to achieve with rigid components. Although CMOS sensor technologies offer robust performance at the millimeter scale, their inherently rigid structures and auxiliary circuits can restrict their suitability for integration with soft microrobots. Similarly, acoustic communication methods typically require rigid resonators, which are incompatible with the mechanical compliance needed for soft microrobots.

In addition, the use of commercial electronic devices with RF signal modulation is constrained by challenges in power delivery, particularly when using smaller RF antennas. Furthermore, to the best of our knowledge, most commercial RF antennas are of similar or larger size than our developed microrobots (with the exception of certain ceramic-based antennas), making them unsuitable for direct integration.

While incorporating microelectronic components may complement or enhance our approach in the future, it lies beyond the scope of the present study, which is primarily focused on the development of fully flexible microrobots. We are grateful for the reviewer's insight and have included a discussion comparing our approach with CMOS-based and acoustic communication solutions to highlight the merit of our approach. We have revised the manuscript accordingly.

Page 4 (lines 63-86)

The integration of these technologies into microrobots became possible in the last few decades due to advances in microfabrication techniques, including those commonly used in microelectromechanical systems (MEMS)^{42, 43, 44} and complementary metal-oxide-semiconductor (CMOS)^{28, 29, 45}, alongside emerging methods such as 3D direct laser writing^{7, 24, 27, 46}. For example, Bandari et al. used microfabrication to create inductively powered microrobots with a strain engineering based form of chemical propulsion²⁹, and Miskin et al. integrated photovoltaics and surface electrochemical actuators for a light powered microrobot²⁸. While advanced fabrication techniques have opened up new possibilities for actuation methods, these implementations still struggle with a critical capability for real practical utility. Namely, these microrobots cannot communicate real-time information about their environment with external systems.

Roboticians have attempted to solve this problem in numerous fashions, whether by incorporating commercial antenna chips (including RFID) or through tethered approaches. These methods, based on readily available, low cost, and robust transmission technologies, can relay important information

about their environment to external agents. For example, Li et al. demonstrated radiofrequency (RF)-capable, magnetically navigated microrobots that wirelessly transmitted temperature and pH data⁴⁷. However, both the chips and required auxiliary circuits force fundamental tradeoffs between signal quality and microrobot size while also requiring a rigid structure. Han and colleagues addressed some of these issues by using microfabrication and compressive buckling techniques to incorporate a flexible electronic sensor array within a balloon catheter that measured pressure and temperature⁴⁸. The soft nature of the flexible system better matches the mechanical properties seen in biology, but the tethered setup still restricts the working range. As such, the question remains whether an untethered can successfully integrate the key aspects of remote navigation, collective behavior, and wireless sensing into a single system while maintaining the key advantage of flexible soft microrobots.

Page 18 (lines 433-439)

As the proposed microrobotic scheme utilizes flexible electronics, it offers advantages over commercial CMOS and RFID devices in terms of flexibility and adaptability, making it safer and more compatible for use in various environments. Acoustic sensing and communication represent a potential alternative;^{65, 66} however, acoustic sensing tends to acquire information over a broad area, making it unsuitable for localized perception. Additionally, existing acoustic communication schemes often rely on rigid implantable structures and are primarily focused on localization, without the ability to externally transmit localized sensory data.

Comment 3: The paper could use more discussion of the role of the environment in communication. 10 GHz RF waves penetrate weakly into water and by extension tissue (see Fig 2, <https://arxiv.org/abs/1306.5709>). Typically, the penetration depth is on the order of a mm. This seems like a major problem: if the best-case signal change for temperature reporting is on the order of a 1-10dB, the same effect could be produced by the device (or transmitter) moving roughly 1 mm (i.e. ~1 body length) deeper in tissue. How would this get de-convolved in practice? Likewise, some discussion of the power needed to operate the device at a given depth, and whether they would be safe, would be helpful. A 1 mm penetration depth would imply that nearly 43 orders of magnitude in power would be lost to reach 10 cm into tissue. How could this be overcome, especially for biomedical applications?

Response:

We are grateful to the reviewer for emphasizing the role of environment in RF communication. While we agree that 10 GHz range have limitations in penetrating water and tissue, we would like to clarify that the signal attenuation we observed, on the order of 1-10 dB, was measured using a vector network analyzer, which operates with relatively low transmission power.

Additionally, we found that phosphate-buffered saline (PBS), which closely mimics the ionic composition of bodily fluids, causes the resonant frequency shift from ~12.0 GHz to ~1.5 GHz. This

shift is beneficial, as lower frequency RF signals exhibit greater penetration depth, enabling reliable differentiation between planar and helical shapes. This benefit of shift to lower resonant frequency is also supported by several studies reporting that RF waves in the 1-5 GHz can be utilized in aqueous and tissue media (i.e., in brain stimulation), demonstrating penetration depths of 16-60 mm (doi.org/10.1098/rsta.2021.0020).

In summary, this challenge can be addressed by increasing the transmission power and operating the microrobots in ion-rich solutions. Our revised manuscript now includes a more explicit discussion of these findings.

Page 12 (lines 281-306)

Investigation of environmental effects

The performance of RF communication-based shape detection can be significantly influenced by environmental factors, including the medium (e.g., ion-containing aqueous solutions) and the surroundings (e.g., experimental setups). First, different media can shift the resonant frequency of RF signals. The basic mechanism of the dipole antenna's resonance is based on the matched wavelength of the electromagnetic wave. The wavelength at the resonant frequency is defined as:

$$\lambda = \frac{c}{f_{r,air}} \quad (5)$$

where c and $f_{r,air}$ are the speed of light and the resonant frequency in air, respectively. In a specific medium, the speed of light v_m is given by:

$$v_m = \frac{c}{n} \quad (6)$$

where n is the refractive index of the medium. Based on Maxwell's electromagnetic field theory, the refractive index is given by $n = \sqrt{\epsilon_r \mu_r}$. For non-magnetic materials, the relative magnetic permeability index μ_r is approximately 1, meaning that the refractive index depends solely on the relative dielectric constant ϵ_r . Thus, the refractive index simplifies to $n = \sqrt{\epsilon_r}$. Since the electromagnetic wavelength for the resonance of dipole antenna needs to remain the same across different media (e.g., air and ion-containing aqueous solutions), Equation (5) and (6) can be rewritten as:

$$\frac{c}{f_{r,air}} = \frac{v_m}{f_{r,medium}} = \frac{c/n}{f_{r,medium}} \quad (7)$$

where $f_{r,medium}$ is the resonant frequency of the dipole antenna in a specific medium. Thus, the resonant frequency $f_{r,medium}$ is defined by Equation (8).

$$f_{r,medium} = \frac{f_{r,air}}{\sqrt{\epsilon_r}} \quad (8)$$

Considering the developed 2 mm × 15 mm microrobots, they exhibited a resonant frequency of ~12 GHz in air. When immersed them in phosphate-buffered saline (PBS, a relative permittivity: ~70), the resonant frequency is expected to shift to ~1.45 GHz. Figure 3h shows that experimentally measured

resonant frequency shifts to ~ 1.52 GHz in the PBS medium, confirming above analytical estimation is reasonable. This analytic estimation and experimental results also aligned with a simulation result, which shows a resonant frequency shifts to ~ 1.47 GHz (Supplementary Fig. 13).

Page 16 (lines 384-391)

We also theoretically and experimentally confirmed that the use of microrobots in the ion-containing aqueous solution medium significantly lowered the resonant frequency and enhanced the RF communication performance, where one-order reduction in resonant frequency enhances the penetration capability of RF signals. Comparing the RF signals of the dipole antenna in PBS at 2 mm and 4 mm depths shows a clearer signal distinction between signals at a greater depth. (Supplementary Fig. 19). This is probably because the increased medium thickness absorbs more RF signals⁶¹, enhancing signal sensitivity. Additionally, since human body fluids are similarly ion-containing aqueous solutions, there is potential applicability for these microrobots in biomedical applications.

Page 18 (lines 422-431)

Meanwhile, studying radio communication in bio-settings reveals both promising aspects and ongoing challenges. For example, the penetration depth of RF electromagnetic waves (1-5 GHz) through tissues falls within a reasonable range (16-60 mm). However, signal reflection at the air-tissue interface remains a limitation, indicating the need for further optimization of external transmitter and receiver coil designs. In addition, while increasing transmitting coil power can help mitigate signal attenuation over these depths, human exposure to RF electromagnetic field should be considered. Although research is ongoing, power levels can be increased within safe limits defined by the tissue heating capacity, measured by the specific absorption rate (SAR), which is generally accepted to be less than 2 W kg^{-1} for localized exposure.⁶⁴ Combined, this wireless communication-capable microrobotic platform holds promise for future medical use.

Figure 3. Design parameter study and shape detecting performance through RF communication. h. RF signal shifts according to shape reconfiguration from helical to planar of seven units of 2×15 microrobots immersed in PBS solution. The resonant frequency of the microrobots in PBS solution shifts to ~ 1.5 GHz, whereas ~ 1.2 GHz in air.

Supplementary Figure 13. Simulation of medium effects on RF communication. The COMSOL simulation with a PBS medium shows a resonant frequency shift to a lower region, consistent with experimental results.

Supplementary Fig. 19. Experimental comparison of microrobots’ RF signal response at the resonant frequency in PBS. As the depth increases, communication performance improves, leading to a more significant absolute amplitude difference.

Comment 4: The device design and fabrication protocol are nice, especially the integration of the magnetic material. This is one of the best parts of the paper, but the extensive discussion of the specifics should be relegated to the supplemental information, in favor of a briefer overview. Additionally, many statements claim standard fabrication techniques are key advances and should be done away with entirely. For instance: “Key advancements included ensuring precise layer alignment”, i.e. using a mask aligner? “Improving adhesion between layers”, i.e. using industry-standard adhesion promoters? “Optimizing the anisotropic layer and IONP pattern orientation”, i.e. magnetizing in the correct direction? “We develop a versatile microfabrication process that incorporates flexible electronics onto a soft microrobotic chassis using combination of thin-film deposition, photolithography, and wet etching techniques”, the versatility of microfabrication is well established. Similar statements appear throughout the manuscript and should be removed.

Response:

We appreciate the reviewer’s positive feedback on our device design and fabrication protocol. In response to suggestions for better clarity, we have revised the main text to present only a concise overview of these methods, while adding the detailed fabrication and integration protocol to the

Supplementary Materials. We have also removed redundant statements describing industry-standard techniques (e.g., mask alignment, adhesion promoters, anisotropic layer orientation) to ensure a focused presentation of our unique integration approach.

Page 5 (lines 88-100)

This research seeks to answer that question with a new microrobotic approach that, as its key innovation, integrates flexible electronics and a shape reconfigurable soft microrobot into a single device. This leverages smart materials to create large, physical changes in the microrobot in response to local stimuli and a flexible electronic antenna design that can take advantage of the entire microrobotic surface to propagate its signal. By coupling the two together, the physical changes to the robotic structure produce equally dramatic changes in the antenna signal character, enabling instant remote communication within a fully flexible microrobotic system. To achieve this, we developed an integration protocol to combine an anisotropic SU-8 passive layer with an iron oxide nanoparticle (IONP) embedded thermally-responsive hydrogel active layer, leading to a temperature dependent helical-to-planar transformation function. We then laminated a flexible dipole antenna for radio communication, completing the microrobot. In this work, we explain the new fabrication route and demonstrate microrobot magnetic navigation, RF communication-based shape detection, localization, and remote temperature sensing. We also show that the collective behavior of multiple microrobots enhances both RF signal sensitivity and reliability.

Page 16 (lines 363-373)

In our manufacturing process, we enhanced the spacer-method-based microfabrication method previously developed by our group⁸ to integrate a flexible dipole antenna into soft microrobotic structures. Specifically, functionalizing the SU-8 surface through techniques like bond cleavage and silanization was crucial for achieving complete shape transformation between the planar and helical states. Since our manufacturing protocols are based on the well-established microfabrication processes, they are compatible with different semiconductor technologies⁵¹, including MEMS^{52, 53} and flexible electronics^{54, 55} processes by combining the electronic elements prior to fabricating the microrobotic structures. This means other electronic devices (e.g., thin-film transistors^{56, 57}, sensors⁵⁸, electroluminescent⁵⁹, and energy harvester⁶⁰) can be used to expand the microrobot's functionality and take advantage of even higher levels of intelligence (e.g., processing, computing). Also, the microfabrication processes are scalable and suitable for mass manufacturing with batch-processing options, producing microrobots of varying sizes and designs.

Page 20 (lines 487-488)

The detailed fabrication protocol flow chart is depicted in the Supplementary Fig. 22 and written in the Supplementary Methods section.

Supplementary Fig. 22. Fabrication and integration workflow chart.

Comment 5: The authors forgo what might be an illuminating discussion about the fact that devices loose motility in high temperature regimes. One could argue that this is a feature, and that the device could be configured to operate in a “seek and destroy” scheme, where they travel around an environment until they reach a region of desired temperature, then planarize and remain their indefinitely. Given the vast expertise of the authors, especially in biomedical applications of microrobots, we’d love to hear them expand on the nuanced ways in which the interweaving of sensing and locomotion may be beneficial.

Response:

We are thankful to the reviewer for this insightful suggestion. Gradient magnetic fields can actuate the microrobots even in their planar state, within the elevated temperature regime. To demonstrate the mobility of planar microrobots, we performed additional experiments.

As mentioned in Response to Comment 1, the shape sensing functionality is important for practical microrobotic applications, including locomotion capability. In biomedical contexts, given the size of the microrobots, we can envision their potential use in diagnostic local temperature sensing

applications within gastrointestinal (GI) tract. In this scenario, the microrobots could be naturally extracted, thus retrieval may not be required.

In addition, to expand the potential of microrobots capabilities in both sensing and locomotion, we have included additional experimental results demonstrating microrobotic localization using a four-by-three array sample holder. We have revised the manuscript accordingly.

Page 2 (lines 25-31, abstract)

As a proof of concept, we present a microrobot, which integrates a thermoresponsive magnetic hydrogel, an anisotropic support structure, and a flexible dipole antenna into a cohesive three-layered design. The microrobot can morph from helical shape at low-temperatures to planar shape at high-temperatures. This shape transformation can be remotely detected by external radio communication receivers, enabling shape-state recognition and environmental temperature sensing. Furthermore, we show that the collective behavior of multiple microrobots enhances the recognition performance by amplifying the signal.

Page 8 (lines 177-180)

Note that microrobots can also be manipulated in dragging motion using gradient fields, which can particularly be useful for controlling planar shape without relying on corkscrew motion capability (see Supplementary Fig. 7 and Movie S4). The average speed of $2\text{ mm} \times 15\text{ mm}$ microrobot was 1.2 mm s^{-1} under a magnetic field of 10 mT and gradient of 120 mT mm^{-1} .

Page 14 (lines 321-332)

Microrobot localization

Wireless communication capability enables microrobot localization and local information perception. Figure 4a illustrated that planar microrobots can be localized through areal scanning using transmitter and receiver antennas. This antenna setup scanned a sample holder divided into a four-by-three array and detected a notable dip (at $\sim 12.2\text{ GHz}$) in the measured RF signal at coordinate position (2, 3), where the planar microrobots were positioned (Fig. 4b; see also Supplementary Fig. 15a).

Additionally, when the microrobots were placed under a cover (Supplementary Fig. 15b,c), the planar and helical microrobots transmitted distinguishable RF signals, enabling shape detection (Fig. 4c). This demonstrates that planar microrobots can be detected via RF communication without the need for a vision system. Finally, these demonstrations imply that developed microrobots can be localized even when covered, allowing their use for a local shape sensing.

Supplementary Fig. 7. Navigation of planar microrobots using a magnetic gradient field. A $2\text{ mm} \times 15\text{ mm}$ planar microrobot (a) and a $2\text{ mm} \times 5\text{ mm}$ planar microrobot (b) were guided by the magnetic dragging force. The magnetic field and gradient were set to 10 mT and 120 mT mm^{-1} , respectively. The average speeds of $2\text{ mm} \times 15\text{ mm}$ and $2\text{ mm} \times 5\text{ mm}$ microrobot were 1.2 mm s^{-1} and 0.7 mm s^{-1} , respectively (Scale bar: 10 mm).

Figure 4. RF communication-based microrobot localization. **a.** Schematic of RF communication-based localization, consisting of a scanning system and a sample holder divided into four-by-three array. The localization system consists of transmitter and receiver coils positioned opposite to each other, with scanning performed along a predefined path. **b.** RF signals across the array exhibit a distinctive signal at position (3,2) corresponding to the location of the planar microrobots. **c.** RF signals from seven units of 2×15 microrobots placed behind a barrier. The clearly distinct signals between helical and planar shapes demonstrate that the microrobots can be detected through the barrier.

Supplementary Fig. 15 Localization setup and shielded experimental setup. a. The experimental platform, divided into a 4×3 array. Two transmission coils were connected to the VNA, and the microrobots were positioned at (3,2). b. A shielded paper cover was placed between the microrobots and the transmitter coil, with an enlarged view shown in (c).

Comment 6: The discussion section could be sharpened. For example: “Previous approaches using smaller antennas faced challenges with inefficient radio communication, unreliable remote sensor powering, and poor signal differentiation in environmental monitoring due to inert sensing mechanisms.” It is not demonstrated in this work that any of these problems were solved. “Our research highlights the microrobots’ potential for biomedical applications requiring operation within the body’s enclosed cavities or conduits.” Only cavities that are large enough to accommodate the size of one or several devices (a few cm as seen in figure 4), and shallow enough that the RF signal can penetrate at order 10 GHz. “This approach could also be extended to develop interactive microrobots that respond to both internal and external stimuli, gather data, communicate with each other, and autonomously manage their functions.” It’s not clear how most of these behaviors will be achieved with the presented platform. If these statements are true, they need clarification and expansion. If not, they should be removed.

Response:

We are thankful to the reviewer for highlighting the need to refine our discussion section. In response, we have carefully revised the text to clarify or remove any statements not directly supported by our findings. We have also narrowed the scope of our conclusions to emphasize the demonstrated functionalities rather than proposing specific microrobotic applications, while explicitly acknowledging the limitations of operating at high RF frequencies. We have revised the manuscript accordingly.

Page 15 (lines 356-449)

We demonstrated remote communication with soft, shape-reconfigurable microrobots, featuring a key innovation in the integration of a flexible dipole antenna. The microrobot's temperature-responsive shape-changing ability directly translates into antenna reconfiguration, providing a clear mechanism for detecting its morphological state remotely. By designing the antenna to match the microrobot's dimensions, we also significantly enhance the sensitivity and reliability of radio communication signals. This approach overcomes challenges in poor signal sensitivity common in previous attempts with similar micro antennas.

In our manufacturing process, we enhanced the spacer-method-based microfabrication method previously developed by our group⁸ to integrate a flexible dipole antenna into soft microrobotic structures. Specifically, functionalizing the SU-8 surface through techniques like bond cleavage and silanization was crucial for achieving complete shape transformation between the planar and helical states. Since our manufacturing protocols are based on the well-established microfabrication processes, they are compatible with different semiconductor technologies⁵¹, including MEMS^{52, 53} and flexible electronics^{54, 55} processes by combining the electronic elements prior to fabricating the microrobotic structures. This means other electronic devices (e.g., thin-film transistors^{56, 57}, sensors⁵⁸, electroluminescent⁵⁹, and energy harvester⁶⁰) can be used to expand the microrobot's functionality and take advantage of even higher levels of intelligence (e.g., processing, computing). Also, the microfabrication processes are scalable and suitable for mass manufacturing with batch-processing options, producing microrobots of varying sizes and designs.

Additionally, we explored various pathways for improving the functionality of the developed microrobots. For example, different antenna designs produced measurably shifted resonant frequencies. Signals from two types of microrobots ($2\text{ mm} \times 15\text{ mm}$ and $2\text{ mm} \times 10\text{ mm}$ with a 10 cm gap between the transmitter and receiver antennas) displayed a difference in resonant frequency shifts of $\sim 300\text{ MHz}$ (Supplementary Fig. 12a(iv) and 12b(iv)). Furthermore, S_{21} varied with different antenna designs. As shown in Supplementary Fig. 17, the square spiral antenna exhibited a resonant frequency of $\sim 20.6\text{ GHz}$. Taken together, it becomes clear that careful antenna design can be used as a tool to distinguish microrobots, which when used in denser distributions, could enhance the signal sensitivity, as shown in Supplementary Fig. 18.

We also theoretically and experimentally confirmed that the use of microrobots in the ion-containing aqueous solution medium significantly lowered the resonant frequency and enhanced the RF communication performance, where one-order reduction in resonant frequency enhances the penetration capability of RF signals. Comparing the RF signals of the dipole antenna in PBS at 2 mm and 4 mm depths shows a clearer signal distinction between signals at a greater depth. (Supplementary

Fig. 19). This is probably because the increased medium thickness absorbs more RF signals⁶¹, enhancing signal sensitivity. Additionally, since human body fluids are similarly ion-containing aqueous solutions, there is potential applicability for these microrobots in biomedical applications.

Continuous tracking of the microrobot's shape demonstrates the feasibility of this approach for real-time, remote monitoring of the microrobot's morphological state. This enables remote recognition of the microrobot's configuration (e.g., helical vs. planar or folded vs. unfolded) during operation, addressing one of the key challenges in the practical use of microrobots. For example, configuration recognition can be advantageous in targeted delivery applications, where active shape reconfiguration (e.g., via magnetic hyperthermia induced by alternating magnetic fields) is required to adapt to complex environments. In such cases, remote communication can inform successful shape transformation, allowing the microrobot to proceed to its next task. Additionally, our microrobot localization experiments demonstrate that the system can reliably detect both position and shape, suggesting its potential for local temperature sensing applications.

The used hydrogels in this study swell at the 34-37 °C of lower critical solution temperature (LCST) with a narrow LCST range⁶², which can be further tuned by adding hydrophilic and hydrophobic monomers and/or modifying the polymer architecture. By manipulating the swelling behavior, the detectable temperature range can be varied⁶³. Based on this, the temperature sensing accuracy, which is originally limited by the intrinsic nature of hydrogel-based temperature sensing, may be further improved through fuzzy logic approach, leveraging the collective effect of microrobots designed with varying phase change temperature. Additionally, the type of hydrogels can be extended to various stimulus-responsive hydrogels (e.g., pH, light, electric, chemical) beyond the heat-detection demonstrated here. Thus, this microrobotic concept has the potential to be expanded into a wide range of devices for monitoring diverse stimuli at the microscale.

Taking the remotely controlled actuation and local environmental sensing together, one begins to see an interesting use case for the developed microrobotic platform in biomedical applications (e.g., binary temperature sensing in localized diagnostic applications. Regions of tumor or inflammation exhibit higher temperatures, typically elevated by 0.5–3 °C). As an initial step, we examined the feasibility of using this microrobot in bio-settings by testing the material biocompatibility with MTT assays (Supplementary Fig. 20). When exposed to human umbilical vein endothelial cell (HUVEC) cultures for 24 and 72 hours, the microrobots demonstrated over 80% of cell viability, indicating acceptable levels of biocompatibility. Meanwhile, studying radio communication in bio-settings reveals both promising aspects and ongoing challenges. For example, the penetration depth of RF electromagnetic waves (1-5 GHz) through tissues falls within a reasonable range (16-60 mm). However, signal reflection at the air-tissue interface remains a limitation, indicating the need for further optimization

of external transmitter and receiver coil designs. In addition, while increasing transmitting coil power can help mitigate signal attenuation over these depths, human exposure to RF electromagnetic field should be considered. Although research is ongoing, power levels can be increased within safe limits defined by the tissue heating capacity, measured by the specific absorption rate (SAR), which is generally accepted to be less than 2 W kg^{-1} for localized exposure.⁶⁴ Combined, this wireless communication-capable microrobotic platform holds promise for future medical use.

As the proposed microrobotic scheme utilizes flexible electronics, it offers advantages over commercial CMOS and RFID devices in terms of flexibility and adaptability, making it safer and more compatible for use in various environments. Acoustic sensing and communication represent a potential alternative;^{65, 66} however, acoustic sensing tends to acquire information over a broad area, making it unsuitable for localized perception. Additionally, existing acoustic communication schemes often rely on rigid implantable structures and are primarily focused on localization, without the ability to externally transmit localized sensory data.

In summary, our proposed microrobot design and manufacturing protocol, which integrates flexible electronics with soft microrobots, enables the development of fully flexible, wireless-sensing microrobots capable of magnetic navigation and collective behavior. The ultra-thin dipole antenna remains fully functional under the mechanical strain induced by shape transformation of the microrobots, transmitting different radio communication signals between planar and helical shapes. The collective behavior further enhances the signal differentiation, improving remote shape detection and ensuring reliability in RF-based temperature sensing. We expect that this advancement could open new possibilities in embodied micro-intelligence that can interact to environment and gather data, thereby paving the way for the autonomous operation of small-scale robots.

Supplementary Fig. 17. Resonant frequency of the new spiral antenna in air. The resonant frequency of the spiral antenna is ~ 20.6 GHz.

Supplementary Fig. 18. Effect of microrobot distribution density on RF communication using COMSOL simulation. A 33% increase in microrobot distribution density resulted in a threefold enhancement of the S_{21} amplitude at the resonant frequency. (Unit area: 1 cm^2)

Supplementary Fig. 19. Experimental comparison of microrobots' RF signal response at the resonant frequency in PBS. As the depth increases, communication performance improves, leading to a more significant absolute amplitude difference.

Responses to Reviewers' Comments

Reviewer 4:

General comments: Generally we feel that the authors have addressed our concerns with their added content. Although we're still unsure about the value of this system vs. other sensing modalities or its practicality in a real world setting, the authors now make factually correct claims and have improved their placement of their work in the broader context of research.

Response:

We sincerely appreciate the reviewer's thorough evaluation of our work and the valuable comments and feedback, which have helped us improve the manuscript and better highlight the novelty of our research. We provide a point-by-point response to address the concerns raised as follows.

Comment 1: It would be to better clarify what's being presented in their new supplemental section on page 24: namely it is unclear if these experiments were performed in air or in water (or a tissue phantom). We recommend the authors spell out these experiments more clearly, especially since if they were done in air, the argument is not very compelling.

Response:

We thank the reviewer for pointing out the unclear description in the supplementary section. We have revised the text to clarify that these experiments were performed in air. In addition, we note that the two-hop relay system can enhance the observed signal strength, as shown in Supplementary Fig. 19. The coupled effects of absorption by the dipole antenna and attenuation in water contribute to a larger signal difference between the two microrobot states. However, as the water depth increases beyond ~4 mm, the overall signal level decreases significantly, which may limit the applicability of this RF concept in such conditions. While we acknowledge these limitations, a more comprehensive investigation of the complex interactions in RF communication, such as attenuation in multi-layered media and interface effects, would be an important topic for future work, though it is beyond the scope of the present study.

Revision:

Overall, the signal variation due to depth is roughly half the magnitude of that caused by morphological changes in air, which can reach to -4 dB (Supplementary Fig. 20).

Comment 2: The power safety argument feels disingenuous. The typical IEEE limit for safe incident power density in RF in the GHz range (surface or whole body) is around $1\text{mW}/\text{cm}^2$ (c.f. <https://ieeexplore.ieee.org/stamp/stamp.jsp?tp=&arnumber=8859679>). If you're emitting 1mW from a

120mm transmitter, it sure seems like you've over the safe exposure limit. Likewise, a comparison to tissue ablation is not reasonable, since in that case you're deliberately trying to produce a thermal effect. This claim should be tempered: state what you power use and put that in the context of the *standard* RF limits for sensing, measurement, and communication.

Response:

We thank the reviewer for raising this important point. The transmitter was powered using a high-frequency VNA, which operates at approximately 1 mW output power. This power is not concentrated within a confined 1 cm² area but instead radiates outward and attenuates rapidly in the medium, resulting in much lower local exposure levels. We would also clarify that this system is intended for operation in localized and restricted environments. According to the references recommended by the reviewer, the local exposure reference level at 10 GHz is approximately 182 mW/cm², which is substantially higher than the levels used in our study. When combined with the additional spreading loss in the medium, this confirms that our operating conditions are well within internationally accepted safety guidelines. We are grateful to the reviewer for us to clarify this point more explicitly.

Revision:

According to the IEEE standard for safety levels with respect to human exposure to electromagnetic fields, the local exposure reference level at 10 GHz is approximately 182 mW cm⁻², which is substantially higher than the levels used in this study⁷¹.

71. *IEEE Standard for Safety Levels with Respect to Human Exposure to Electric, Magnetic, and Electromagnetic Fields, 0 Hz to 300 GHz. In: IEEE Std C95.1-2019 (Revision of IEEE Std C95.1-2005/ Incorporates IEEE Std C95.1-2019/Cor 1-2019)) (2019).*